# Direct Advantage Estimation in Partially Observable Environments

## Abstract

Direct Advantage Estimation (DAE) was recently shown to improve sample-efficiency of deep reinforcement learning algorithms. However, DAE assumes full observability of the environment, which may be restrictive in realistic settings. In the present work, we first show that DAE can be extended to partially observable domains with minor modifications. Secondly, we address the increased computational cost due to the need to approximate the transition probabilities through the use of discrete latent dynamics models. Finally, we empirically evaluate the proposed method using the Arcade Learning Environments, and show that it is scalable and sample-efficient.

## 1 Introduction

Real-world decision-making problems often involve incomplete information, where observations received by the agents are not enough to fully determine the underlying state of the system. For example, a robot navigating a building may only have a local view of its surroundings; a doctor has to decide the course of treatment for a patient based on a limited set of test results. The Partially Observable Markov Decision Process (POMDP) framework (Kaelbling et al., 1998) provides a generalization of the fully observable MDP framework (Puterman, 2014) to tackle these problems.

While reinforcement learning (RL) (Sutton and Barto, 2018) paired with deep neural networks (deep RL) has achieved unprecedented results in various domains (Mnih et al., 2015; Berner et al., 2019; Schrittwieser et al., 2020; Ouyang et al., 2022; Wurman et al., 2022), it is known to be challenging to train and often requires millions or billions of samples (Henderson et al., 2018). Approximating the state(-action) value functions ($Q^\pi$ or $V^\pi$) is a crucial part of training deep RL agents. However, these functions often depend strongly on the policy, making them highly non-stationary and difficult to learn. Recently, Pan et al. (2022) demonstrated that the advantage function is more stable under policy variations, proposing Direct Advantage Estimation (DAE) to learn the advantage function directly for on-policy settings. DAE demonstrated strong empirical performance, but is restricted to on-policy settings. Later, Pan and Schölkopf (2024) observed that the return of a trajectory can be decomposed into two different advantage functions, which enabled a natural generalization of DAE to off-policy settings (Off-policy DAE). Off-policy DAE was reported to further improve the sample efficiency of DAE; however, the method suffers from significantly increased computational complexity due to the need to learn a high dimensional generative model to approximate the transition probabilities.

The present work explores the feasibility of DAE in partially observable domains, and ways to reduce its computational complexity. More specifically, the contributions are:

- We show that Off-policy DAE can be applied to POMDPs with minor modifications to the constraints of the objective.
- We address the problem of increased computational cost of Off-policy DAE by modeling transitions in a low dimensional embedding space, which circumvents the need to model high dimensional observations.
- We show that truncating trajectories, a common technique for reducing the computational cost of training POMDP agents, can lead to confounding and degrade performance.
- We evaluate our method empirically using the Arcade Learning Environment (Bellemare et al., 2013), and perform an extensive ablation study to show the contributions of various corrections and hyperparameters.

## 2 BACKGROUND

In the present work, we consider a discounted POMDP defined by the tuple $(\mathcal{S}, \mathcal{A}, T, \Omega, \mathcal{O}, r, \gamma)$ (Kaelbling et al., 1998), where $\mathcal{S}$ is the state space, $\mathcal{A}$ is the action space, $T(s, a, s')$ denotes the transition probability from state $s$ into state $s'$ after taking action $a$, $\Omega$ is the observation space, $\mathcal{O}(s, o)$ denotes the probability of observing $o \in \Omega$ in state $s$, $r(s, a)$ denotes the reward received by the agent after taking action $a$ in state $s$, and $\gamma \in [0, 1)$ denotes the discount factor. When the context is clear, we shall simply denote $T(s, a, s')$ by $p(s'|s, a)$, and $\mathcal{O}(s, o)$ by $p(o|s)$. In this work, we shall consider the case where $\mathcal{S}$, $\mathcal{A}$, and $\Omega$ are finite. An agent in a POMDP cannot directly observe the states, but only the observations emitted from the state through $\mathcal{O}$. We consider the infinite-horizon discounted setting, where the goal of an agent is to find a policy $\pi$, which maximizes the expected cumulative reward, i.e., $J(\pi) = \mathbb{E}_\pi \left[ \sum_{t=0}^{\infty} \gamma^t r(s_t, a_t) \right]$.

In fully observable environments, one can estimate the state(-action) value function $V^\pi(s)$ (or $Q^\pi(s, a)$) as the states are observed directly. In POMDPs, however, agents do not observe states directly, and have to estimate the values based on the observed history (information vector) $h_t = (o_0, a_0, r_0, o_1, ..., o_t)$ (Bertsekas, 2012). Similar to their counterparts in MDPs, we can define the value functions by:

$$V^\pi(h_t) = \mathbb{E}_\pi \left[ \sum_{t'=0}^{\infty} \gamma^{t'} r_{t+t'} \middle| h_t \right], \quad Q^\pi(h_t, a_t) = \mathbb{E}_\pi \left[ \sum_{t'=0}^{\infty} \gamma^{t'} r_{t+t'} \middle| h_t, a_t \right]. \quad (1)$$

### 2.1 DIRECT ADVANTAGE ESTIMATION

Aside from $Q$ and $V$, another function of interest is the advantage function defined by $A^\pi(s, a) = Q^\pi(s, a) - V^\pi(s)$ (Baird, 1995). Recently, Pan et al. (2022) proposed Direct Advantage Estimation (DAE) to estimate the advantage function by minimizing

$$\mathcal{L}(\hat{A}, \hat{V}) = \mathbb{E}_\pi \left[ \left( \sum_{t=0}^{n-1} \gamma^t (r_t - \hat{A}_t) + \gamma^n \hat{V}_{\text{target}}(s_n) - \hat{V}(s_0) \right)^2 \right] \quad \text{s.t.} \sum_{a \in \mathcal{A}} \hat{A}(s, a)\pi(a|s) = 0, \quad (2)$$

where $\hat{V}_{\text{target}}$ is a given bootstrapping target, $r_t = r(s_t, a_t)$, and $\hat{A}_t = \hat{A}(s_t, a_t)$. The constraint ensures the centering property of the advantage function (i.e., $\mathbb{E}_\pi[A^\pi(s, a)|s] = 0$). The minimizer of $\mathcal{L}(\hat{A}, \hat{V})$ can be viewed as a multi-step estimate of $(A^\pi, V^\pi)$. One limitation of DAE is that it is on-policy, that is, the behavior policy ($\mathbb{E}_\pi$) has to be the same as the target policy ($\pi$ in the constraint).

Pan and Schölkopf (2024) extended DAE to off-policy settings (Off-policy DAE), by showing that if we view stochastic transitions as actions from an imaginary agent (nature), then the return of a trajectory can be decomposed using the advantage functions from both agents as

$$\sum_{t=0}^{\infty} \gamma^t r(s_t, a_t) = \sum_{t=0}^{\infty} \gamma^t \left( A^\pi(s_t, a_t) + B^\pi(s_t, a_t, s_{t+1}) \right) + V^\pi(s_0), \quad (3)$$

where $B^\pi(s_t, a_t, s_{t+1}) = \gamma V^\pi(s_{t+1}) - \gamma \mathbb{E}_{s' \sim p(\cdot|s_t, a_t)}[V^\pi(s')|s_t, a_t]$ is the advantage function of nature, which was referred to as *luck* as it quantifies how much of the return is caused by nature. This decomposition admits a natural generalization of DAE into off-policy settings by incorporating $\hat{B}$ into the objective function (Equation 2):

$$\mathcal{L}(\hat{A}, \hat{B}, \hat{V}) = \mathbb{E}_\mu \left[ \left( \sum_{t=0}^{n-1} \gamma^t (r_t - \hat{A}_t - \hat{B}_t) + \gamma^n \hat{V}(s_n) - \hat{V}(s_0) \right)^2 \right]$$

$$\text{subject to} \quad \begin{cases} \mathbb{E}_{a \sim \pi(\cdot|s)}[\hat{A}(s, a)] = 0 \\ \mathbb{E}_{s' \sim p(\cdot|s, a)}[\hat{B}(s, a, s')] = 0 \end{cases}. \quad (4)$$

Contrary to Equation 2, the behavior policy ($\mathbb{E}_\mu$) and the target policy ($\pi$ in the constraint) need not be equal. Intuitively, $\hat{A}$ and $\hat{B}$ can be viewed as corrections for stochasticity originating from the policy and the transitions, respectively. Under mild assumptions, one can show that $(A^\pi, B^\pi, V^\pi)$ is

the unique minimizer of this objective function, suggesting that we can perform off-policy policy evaluation by minimizing the empirical version of this objective function. However, Off-policy DAE has some limitations:

- The method assumes fully observable MDPs, which can be restrictive in realistic settings.

- Enforcing the $\hat{B}$ constraint in Equation 4 requires estimating the transition probability $p(s'|s, a)$, which can be difficult when the state space is high-dimensional (e.g., images). Pan and Schölkopf (2024) reported that learning the transition probability can drastically increase the computational complexity ($\sim$7 fold increase in runtime).

We address these issues in Section 3.

## 3  RETURN DECOMPOSITION IN POMDPs

The key observation of Pan and Schölkopf (2024) is that the return can be decomposed using two different advantage functions (Equation 3). Here, we show that such a decomposition also exists in POMDPs.

Firstly, we can define the advantage function in POMDPs by

$$A^\pi(h_t, a_t) = Q^\pi(h_t, a_t) - V^\pi(h_t) = \mathbb{E}_\pi \left[ \sum_{t'=0}^\infty \gamma^{t'} r_{t+t'} \bigg| h_t, a_t \right] - \mathbb{E}_\pi \left[ \sum_{t'=0}^\infty \gamma^{t'} r_{t+t'} \bigg| h_t \right]. \quad (5)$$

Similar to its counterpart in MDPs, this function also satisfies the centering property, namely $\sum_{a \in \mathcal{A}} \pi(a_t|h_t) A^\pi(h_t, a_t) = 0$. The next question is how we can similarly define the luck function $B^\pi$ such that the return can be decomposed, and whether this function also satisfies the centering condition.

We proceed by examining the difference between the return and the sum of the advantage function along a given trajectory $(o_0, a_0, r_0, o_1, a_1, r_1, ...)$

$$\sum_{t=0}^\infty \gamma^t r_t - \left( \sum_{t=0}^\infty \gamma^t A^\pi(h_t, a_t) + V^\pi(h_0) \right) = \sum_{t=0}^\infty \gamma^t \left( r_t + \gamma V^\pi(h_{t+1}) - Q^\pi(h_t, a_t) \right). \quad (6)$$

This equation suggests that we can define the luck function as

$$B^\pi(h_t, a_t, h_{t+1}) = r_t + \gamma V^\pi(h_{t+1}) - Q^\pi(h_t, a_t) \quad (7)$$

Remember that $h_{t+1}$ is simply the concatenation of $h_t$ and $(a_t, r_t, o_{t+1})$, meaning that we can rewrite $B^\pi(h_t, a_t, h_{t+1})$ as $B^\pi(h_t, a_t, r_t, o_{t+1})$. We thus see that the $B^\pi$ defined this way also satisfies a (slightly different) centering property, namely,

$$\mathbb{E}_{(r_t, o_{t+1}) \sim p(\cdot|h_t, a_t)} \left[ B^\pi(h_t, a_t, r_t, o_{t+1}) | h_t, a_t \right] = 0. \quad (8)$$

Essentially, this equation differs from its MDP counterpart by the variables that are being integrated. In POMDPs, since the agent does not observe the underlying state, we simply integrate the variables being observed after taking an action (i.e., the immediate reward and the next observation).

Finally, we arrive at the following generalization:

**Proposition 1** (Off-policy DAE for POMDPs). *Given behavior policy $\mu$, target policy $\pi$, and backup length $n \geq 0$. $(A^\pi, B^\pi, V^\pi)$ is a minimizer of*

$$\mathcal{L}(\hat{A}, \hat{B}, \hat{V}) = \mathbb{E}_\mu \left[ \left( \sum_{t'=0}^{n-1} \gamma^{t'} \left( r_{t+t'} - \hat{A}_{t+t'} - \hat{B}_{t+t'} \right) + \gamma^n \hat{V}(h_{n+t}) - \hat{V}(h_t) \right)^2 \right] \quad (9)$$

$$\text{subject to} \begin{cases} \mathbb{E}_{a \sim \pi(\cdot|h)}[\hat{A}(h, a)|h] = 0 & \forall h \in \mathcal{H} \\ \mathbb{E}_{(r, o') \sim p(\cdot|h, a)}[\hat{B}(h, a, r, o')|h, a] = 0 & \forall (h, a) \in \mathcal{H} \times \mathcal{A} \end{cases},$$

*where $\mathcal{H}$ is the set of all trajectories of the form $(o_0, a_0, r_0, ...o_t)$, $\hat{A}_t = \hat{A}(h_t, a_t)$, and $\hat{B}_t = \hat{B}(h_t, a_t, r_t, o_{t+1})$. Furthermore, the minimizer is unique if, for any trajectory $h \in \mathcal{H}$, $p_\mu(h) > 0$.*

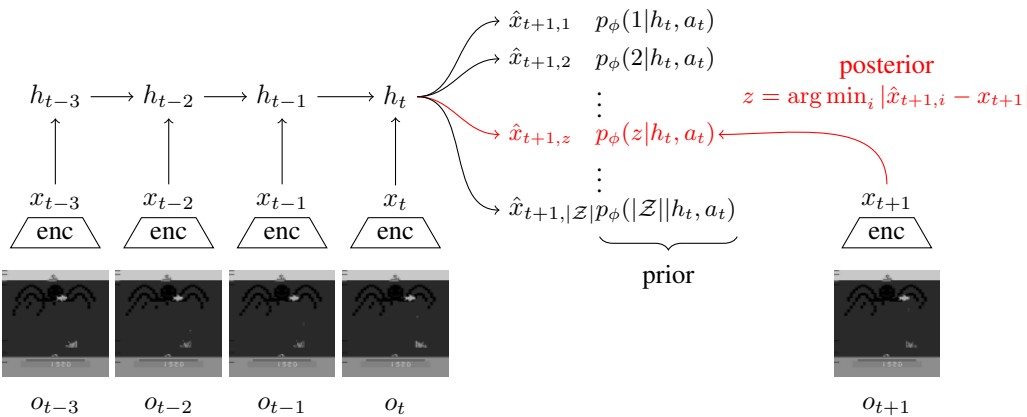

Figure 1: The latent dynamics model first embeds observations ($o_t$) into low dimensional vectors ($x_t$), which are then processed by an RNN to capture the information vectors $h_t$ (for illustrative purpose, we omit conditioning on previous actions and rewards). At each time-step, $|\mathcal{Z}|$ predictions of the embeddings $\hat{x}_{t+1,\cdot}$ along with their probabilities $p_\phi(\cdot|h_t, a_t)$ are generated to capture the distribution of $x_{t+1}$. During training, gradients only propagate through one of the predictions determined by the posterior, which, in this case, is a one-hot vector.

See Appendix A for a proof. Again, we remind the reader that Proposition 1 differs from its MDP counterpart (Equation 4) by simply replacing states with histories, and transition probabilities with conditional densities of the observed variables (in the $\hat{B}$ constraint). This is a consequence of the fact that POMDPs can be reformulated as MDPs using information vectors (Bertsekas, 2012). Like DAE, this can be seen as an off-policy multi-step method for value approximation, as the objective function includes $n$-step rewards. Deploying this method in practice, however, can be computationally heavy, and we discuss methods to reduce its computational complexity below.

### 3.1 PRACTICAL CONSIDERATIONS — ENFORCING CONSTRAINTS

We first discuss how to (approximately) enforce the two centering constraints in the objective function (Equation 9) by reparameterizing the function approximators.

The $\hat{A}$ constraint can be easily enforced upon a given function approximator $\hat{f}(h, a)$ for a given policy $\pi$ by constructing $\hat{A}(h, a) = \hat{f}(h, a) - \sum_{a \in \mathcal{A}} \hat{f}(h, a)\pi(a|h)$ (Wang et al., 2016). The $\hat{B}$ constraint, on the other hand, is much more challenging, as it requires knowledge of the transition probabilities $p(\cdot|h, a)$. In the original Off-policy DAE implementation (Pan and Schölkopf, 2024), this was achieved by encoding transitions $(h, a, r, o')$ into a small discrete latent space $z \in \mathcal{Z}$ using a conditional variational autoencoder (Kingma and Welling, 2013; Sohn et al., 2015), and constructing $\hat{B}(h, a, r, o')$ from a given function approximator $\hat{g}(h, a, z)$ by

$$\hat{B}(h, a, r, o') = \mathbb{E}_{z \sim q_\phi(\cdot|h,a,r,o')}[\hat{g}(h, a, z)|h, a, r, o'] - \mathbb{E}_{z \sim p_\phi(\cdot|h,a)}[\hat{g}(h, a, z)|h, a], \quad (10)$$

where $q_\phi(\cdot|h, a, r, o')$ is the approximated posterior (encoder), $p_\phi(\cdot|h, a)$ is the prior, and $\phi$ is the parameters of the CVAE[1]. In practice, the expectations can be computed efficiently since $\mathcal{Z}$ is discrete. It then follows that $\mathbb{E}_{(r,o') \sim p(\cdot|h,a)}[\hat{B}(h, a, r, o')|h, a] \approx 0$. This approach, however, can be computationally heavy if observations are high dimensional due to the need to reconstruct observations.

To reduce computational complexity, we propose to learn a discrete dynamics model purely in the embedding space[2] (see Figure 1). This is achieved by first embedding observations into a low dimensional vector $x = \texttt{enc}(o) \in \mathbb{R}^d$ (with $d \ll \dim(\Omega)$), where enc denotes the encoder (e.g., a

---

[1] We adapt the POMDP setting here for clarity, but note that this was originally developed for $\hat{B}(s, a, s')$.

[2] We will refer to the space of encoded observations as the embedding space, and $\mathcal{Z}$ as the latent space of the CVAE to avoid confusion.

convolutional network), and learning to predict $x_{t+1} = \texttt{enc}(o_{t+1})$ from the observed history $(h_t, a_t)$. This approach is similar to the self-predictive representation (SPR) (Schwarzer et al., 2020); however, SPR only produces a single prediction, which cannot capture the stochasticity of transitions. We address this by combining SPR with the Winner-Takes-All (WTA) loss (Lee et al., 2015; Guzman-Rivera et al., 2012), which was shown to be useful for modeling stochastic predictions. More specifically, we combine them by: (1) making $|\mathcal{Z}|$ predictions of the next embedding (note that $|\mathcal{Z}|$ is an integer since we are using discrete latent variables), and (2) minimizing only the best prediction. More specifically, the objective function is:

$$\mathcal{L}_{\text{rec}} = \sum_{z \in \mathcal{Z}} w(\hat{x}_{t+1,z}, x_{t+1})|\hat{x}_{t+1,z} - \texttt{sg}(x_{t+1})|^2, \tag{11}$$

$$w(\hat{x}_{t+1,z}, x_{t+1}) = \begin{cases} 1, & z = \arg\min_i |\hat{x}_{t+1,i} - x_{t+1}| \\ 0, & \text{otherwise} \end{cases}, \tag{12}$$

where $\texttt{sg}$ denotes stop-gradient. Intuitively, this can be seen as performing $k$-means clustering (with $k = |\mathcal{Z}|$) in the embedding space with centroids $\hat{x}_{\cdot,z}$ (Rupprecht et al., 2017). Next, note that, the objective (Equation 11) is equivalent to a conditional vector-quantized VAE (VQ-VAE) (Van Den Oord et al., 2017), with posterior $q_\phi(z|h_t, a_t, x_{t+1}) = w(\hat{x}_{t+1,z}(h_t, a_t), x_{t+1})$, and codebook $\hat{x}_{t+1,z}(h_t, a_t)$ that are dependent on the information vector $h_t$. Consequently, we can learn the prior by minimizing the KL-divergence between the prior $p_\phi(z|h_t, a_t)$ and the posterior $q_\phi(z|h_t, a_t, x_{t+1})$. Once we learn a conditional VQ-VAE, we can approximate the $\hat{B}$ constraint using Equation 10. The constraint in the objective (Equation 9) indicates that we should also consider stochasticity of the rewards, which can be similarly achieved by making multiple reward predictions and adding a reward reconstruction term to the objective function.

In practice, we found that using shallow MLPs to model the dynamics already achieves strong empirical performance with negligible computational cost compared to other parts of the system. In addition, we found it possible to learn the RL objective (Equation 9) and the dynamics model jointly end-to-end to further reduce computational complexity compared to learning them separately as done by Pan and Schölkopf (2024).

## 3.2 Practical Considerations — Truncating Sequences & Confounding

As states are now replaced by histories, we have to process sequences of observations instead of singular states. In modern deep RL, this is typically achieved by using recurrent neural networks (RNNs), such as LSTMs or GRUs (Hochreiter and Schmidhuber, 1997; Hausknecht and Stone, 2015; Mnih et al., 2016; Kapturowski et al., 2018; Gruslys et al., 2018; Cho et al., 2014; Hafner et al., 2023). This can be computationally heavy during training when trajectories extend to thousands of steps. Instead, it is common to truncate histories by sampling random segments of trajectories from the replay buffer, and initialize the hidden recurrent state with the first few steps (burn-in) before updating the values (Kapturowski et al., 2018), as demonstrated below:

$$\Big(\underbrace{o_0, a_0, r_0, \cdots, o_{t-k-1}}_{\text{truncated}}, \underbrace{a_{t-k-1}, r_{t-k-1}, o_{t-k}, \cdots}_{\text{burn in (initialize RNN state)}}, \underbrace{o_t, a_t, r_t, \cdots, o_{t+n}}_{\text{value updates}}\Big)$$

Here, we highlight a problem that is commonly overlooked due to confounding (Pearl, 2009). Confounding can bias our estimates when the unobserved variables (the truncated part of the trajectory $h_{0:t-k-1}$) simultaneously influence both the input variables (the remaining part of the trajectory $h_{t-k:t}$) and the output variables (the future rewards $r_{t'}$ for $t' \geq t$). We illustrate this problem with a toy example (see Figure 2). In this environment, the optimal policy is $\pi^*(\text{up}|o_0) = p \in [0, 1]$ (arbitrary), and $\pi^*(a|o_0, a_0=a, o_1) = 1$ (repeat previous actions). Now, let us consider the case where the behavior policy is the optimal policy with $\pi^*(\text{up}|o_0) = 0.5$, but the truncation length is 0 (i.e., memoryless) for the target policy. In this case, we will incorrectly infer that $V^\pi(o_1) = Q^\pi(o_1, \cdot) = 1$ for any target policy $\pi$, since all the collected trajectories receive a reward 1 irrespective of the action $a_1$. This is a classic example of confounding, where the unobserved variable $a_0$ behaves as a confounder that biases our estimates.

In the online setting, we can partially eliminate confounding by conditioning both the behavior policy and the target policy on the same set of variables (i.e., same memory capacity). This then breaks the

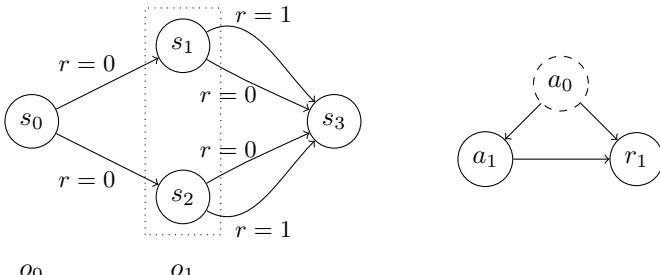

Figure 2: **Left**: A toy POMDP with 4 states and 2 actions. The nodes and the arrows represent the states and the actions (up, down), respectively. $s_0$ is the starting state and $s_3$ is the terminal (absorbing) state. The agent does not observe the underlying state but only the emitted observation at each time step, $o_0$ and $o_1$, where both $s_1$ and $s_2$ emit the same observation $o_1$. **Right**: The causal relationship between $a_0$, $a_1$, and $r_1$. We ignore other variables as they do not influence $r_1$. The variable $a_0$ can act as a confounder during training when the target policy is memoryless.

causal influence from $h_{0:t-k-1}$ to $a_{t'}$ for all $t' \geq t$. While this does not fully eliminate confounding since $h_{t-k:t}$ and $r_t$ can still be influenced by $h_{0:t-k-1}$, we empirically show that this simple change can have non-trivial effects on the agent's performance (see Section 4). Finally, we remind the reader that this is not a limitation of the proposed method, but a common issue due to partial observability.

## 4 EXPERIMENTS

We examine the performance of the proposed method using the Arcade Learning Environment (ALE) (Bellemare et al., 2013), which includes environments with diverse dynamics and various degrees of partial observability. More specifically, we use the five environments subset (Battle Zone, Double Dunk, Name This Game, Phoenix, Qbert) suggested by Aitchison et al. (2023) because it was found that the learning performance in this subset strongly correlates with the overall performance of an algorithm. We use the same environment setting as the Dopamine baselines (Castro et al., 2018), which largely follows the protocols proposed by Machado et al. (2018), including the use of sticky actions (repeat previous action with a certain probability) and discarding end-of-life signals. Note that while sticky actions were originally proposed to inject stochasticity into the environments, they also introduce additional partial observability due to its dependency on previous actions.

We evaluate our method using a DQN-like (Mnih et al., 2015) agent with some modifications, which we briefly summarize below. (1) **Recurrent Architecture**: We do not use frame-stacking, but simply use an LSTM after the convolutional encoder to process sequences of observations. Aside from image inputs, we also feed previous actions and rewards into the LSTM. (2) **DAE objective**: We replace the 1-step Q-learning objective with our multi-step DAE objective (Equation 9), and use three separate MLPs on top of the LSTM to model $\hat{A}$, $\hat{B}$, and $\hat{V}$. (3) **Discrete Latent Dynamics Model**: We use three additional MLPs on top of the LSTM to estimate the next observation embedding $\hat{x}_{t+1}(h_t, a_t, z_t)$, the immediate reward $p(r_t|h_t, a_t)$, and the prior probability $p(z_t|h_t, a_t)$. (4) **Exponential Moving Average**: Similar to SPR, we use an exponential moving average of the online network as the target network to generate the next observation embeddings for the dynamics model. This target network is also used to construct smoothly changing target policies and value bootstrapping targets for the DAE objective. (5) **Deeper Network**: We use the deep residual network proposed by Espeholt et al. (2018) instead of the shallow three-layer convolutional network, which we found to enjoy better scalability and improved sample efficiency. In our implementation, we use a CNN-LSTM backbone with MLP heads on top to model the value functions $(\hat{A}, \hat{B}, \hat{V})$ and the dynamics $(\hat{x}, p(\cdot|h, a))$. For more details, we refer the reader to Appendix C.

In the following experiments, we train the agents for 20 million frames (5 million environment steps due to frame-skipping), and evaluate the agent every 1 million frames by averaging the cumulative scores of 50 episodes.

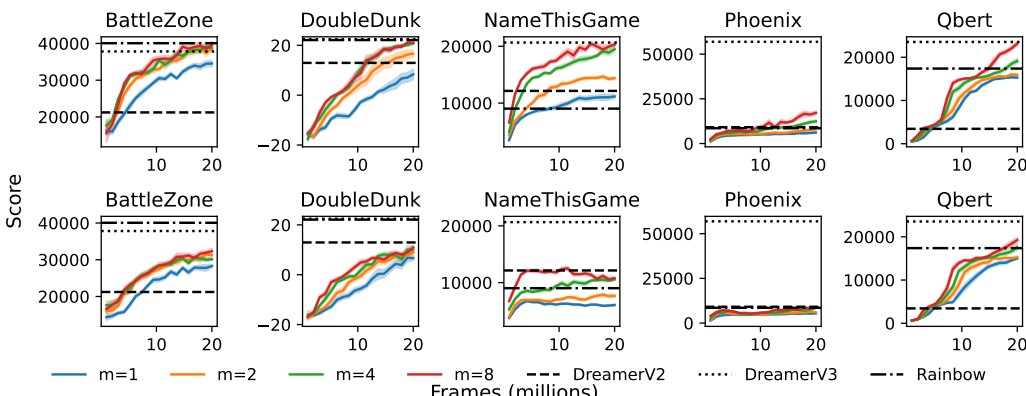

Figure 3: Comparing scalability and sample efficiency with off-policy correction (top row) and without it (bottom row). Results are aggregated over 10 random seeds. Lines and shadings represent the mean and 1 standard error, respectively. $m$: width multiplier of the convolutional layers.

Table 1: Effect of model capacity and off-policy correction on the final evaluation score. Scores were aggregated over 10 random seeds after 20M training frames. Values represent (mean)±(1 standard error). $O$: Off-policy correction. $m$: width multiplier. RBW: Rainbow (200M frames). DV2: DreamerV2 (20M frames). DV3: DreamerV3 (20M frames).

| $O$ | $m$ | BattleZone | DoubleDunk | NameThisGame | Phoenix | Qbert |
|---|---|---|---|---|---|---|
| | 1 | $35044 \pm 986$ | $8.80 \pm 2.13$ | $10977 \pm 441$ | $5666 \pm 54$ | $15313 \pm 56$ |
| ✓ | 2 | $38164 \pm 603$ | $17.68 \pm 1.15$ | $14308 \pm 289$ | $8010 \pm 270$ | $15831 \pm 146$ |
| | 4 | $40098 \pm 900$ | $21.17 \pm 0.45$ | $18682 \pm 580$ | $13593 \pm 1036$ | $19697 \pm 599$ |
| | 8 | $39262 \pm 763$ | $21.49 \pm 0.38$ | $19638 \pm 633$ | $18127 \pm 1075$ | $23451 \pm 415$ |
| | 1 | $27700 \pm 433$ | $7.59 \pm 1.55$ | $6066 \pm 74$ | $5366 \pm 94$ | $14944 \pm 124$ |
| ✗ | 2 | $31310 \pm 554$ | $11.00 \pm 0.95$ | $8090 \pm 246$ | $5835 \pm 189$ | $15599 \pm 239$ |
| | 4 | $30600 \pm 490$ | $11.16 \pm 1.22$ | $10171 \pm 270$ | $8838 \pm 515$ | $17529 \pm 794$ |
| | 8 | $32438 \pm 752$ | $11.46 \pm 0.90$ | $11007 \pm 186$ | $9641 \pm 408$ | $19868 \pm 732$ |
| RBW | | $40061 \pm 1866$ | $22.12 \pm 0.34$ | $9026 \pm 193$ | $8545 \pm 1286$ | $17383 \pm 543$ |
| DV2 | | $21225 \pm 743$ | $12.95 \pm 1.31$ | $12145 \pm 87$ | $9117 \pm 2151$ | $3441 \pm 969$ |
| DV3 | | $37818 \pm 3979$ | $22.46 \pm 0.35$ | $20652 \pm 1002$ | $56889 \pm 3402$ | $23540 \pm 1976$ |

**Off-policy correction is crucial for scaling** In value-based deep RL, it is common to use biased multistep value targets by ignoring off-policy corrections (Hessel et al., 2018; Hernandez-Garcia and Sutton, 2019; Kapturowski et al., 2018; Horgan et al., 2018). However, as we will demonstrate, ignoring off-policy corrections can hurt the agent's scalability and final performance. Following Pan and Schölkopf (2024), we partially disable off-policy corrections by setting $\hat{B} \equiv 0$, which can be seen as ignoring corrections for the stochasticity of the environment (this is due to $B^\pi \equiv 0$ for arbitrary $\pi$ if the environment is deterministic[3]). We also remind the reader that the widely used biased $n$-step method is more aggressive and equivalent to enforcing both $\hat{A} \equiv 0$ and $\hat{B} \equiv 0$. For scaling, we simply multiply the width of the convolutional layers in the encoder by a multiplier $m$. As a comparison, we include both model-based (DreamerV2 (Hafner et al., 2020) and DreamerV3 (Hafner et al., 2023)), and model-free (Rainbow (Castro et al., 2018)) algorithms as baselines. For DreamerV2 and DreamerV3, we report their scores evaluated at 20 million training frames as their model capacities ($\sim$ 20M and $\sim$ 200M parameters, respectively) are similar to ours (the $m = 8$ model has ~50M parameters), but also note that both Dreamer methods were originally trained for 200M frames. For Rainbow, we simply report its score at 200 million training frames, since it uses a much smaller model. Figure 3 and Table 1 summarize the results. Firstly, we find that scaling up can substantially improve the sample efficiency of our method, and we see efficiency comparable to DreamerV3 in 3

---

[3]Based on the observed transition probability $p(r, o'|h, a)$ instead of the state transition probability $p(s'|s, a)$.

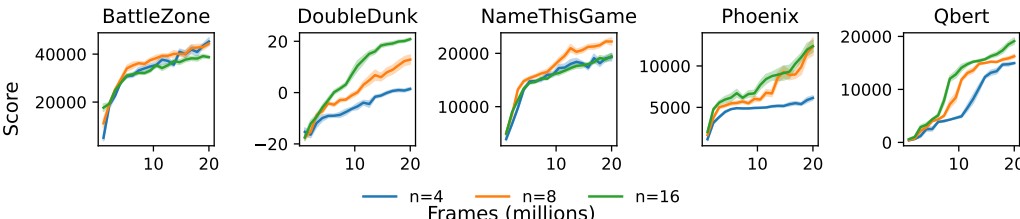

Figure 4: Effect of backup length $n$. Results are aggregated over 10 random seeds. Lines and shadings represent the mean and 1 standard error, respectively.

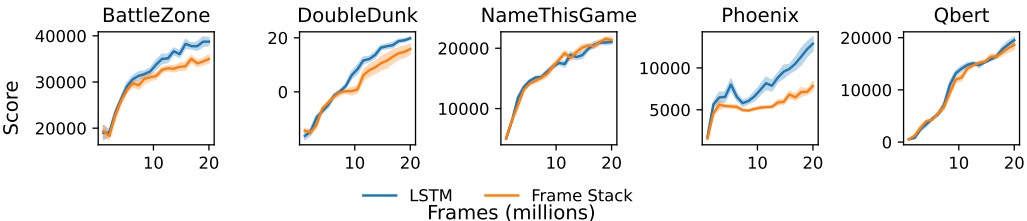

Figure 5: Comparing LSTM to frame-stacking. Results are aggregated over 10 random seeds. Lines and shadings represent the mean and 1 standard error, respectively.

out of 5 environments with our $m = 8$ model, and better efficiency compared to DreamerV2 in most environments with our smaller model $m = 2$. Comparisons with Rainbow also demonstrate that our method can achieve similar performance while using only 10% of the training frames. Secondly, we see that disabling off-policy corrections can drastically degrade the performance and limit the benefit of scaling. These results also suggest that the learned latent dynamics model can indeed capture the stochasticity of the environments, as approximating the $\hat{B}$ constraint hinges on the dynamics model.

Next, we perform ablation studies to better understand the contribution of each part. For the following experiments, we use the $m = 4$ model to reduce the computational cost.

**Effect of backup length.** Multi-step learning allow reward information to propagate faster and reduce dependencies on the bootstrapping target, and it was found to stabilize and speedup training (Hernandez-Garcia and Sutton, 2019; Van Hasselt et al., 2018). However, it can also increase the variance of value updates, and choosing the backup length $n$ can be seen as a bias-variance tradeoff (Kearns and Singh, 2000). Figure 4 summarizes the effect of $n$ for our DAE agent. In general, we find using larger $n$ to be beneficial, except for Battle Zone and Name This Game, where we see that increasing the backup length beyond 8 can hurt the performance.

**Frame-stacking can be suboptimal.** Frame-stacking has been the standard approach to approximate the ALE environments as MDPs since its introduction by Mnih et al. (2015). Here, we examine the effect of our proposed POMDP correction compared to approximating the environment as an MDP via frame-stacking.[4] This can also be seen as a comparison between the POMDP version of DAE and its MDP counterpart. For fair comparison, we set the truncation length of the LSTM agent to 4 (this also applies to action selection), such that both agents have the same context length during action selection, and differ only in how the values are learned. In Figure 5, we see the LSTM agent to perform at least on par with the frame-stacking agent, while being significantly better in three of the environments. This indicates that our POMDP correction is indeed effective when the underlying environments are POMDPs.

**Confounding can degrade performance.** As pointed out in Section 3.2, truncating sequences is essential to reducing computational cost, but naively truncating sequences without adjusting the behavior policy can lead to bias in value estimations due to confounders. Here, we test the impact of truncation length and the confounding bias in the ALE. To test this, we compare two different

---

[4]For easier comparison, we use a different frame-stacking implementation. See Appendix C.4 for details.

Table 2: Effect of confounding and truncation length on the final evaluation score. Scores were aggregated over 10 random seeds after 20M training frames. Values represent (mean)±(1 standard error). $k$: truncation length. R: recurrent behavior policy. diff: relative difference of the score.

| $k$ | R | BattleZone | DoubleDunk | NameThisGame | Phoenix | Qbert |
|---|---|---|---|---|---|---|
| 4 | ✗ | $39404 \pm 899$ | $19.88 \pm 0.58$ | $21283 \pm 412$ | $12945 \pm 590$ | $19825 \pm 559$ |
| | ✓ | $36762 \pm 887$ | $17.67 \pm 1.43$ | $19760 \pm 800$ | $12234 \pm 379$ | $18718 \pm 493$ |
| diff(%) | | $-6.70\%$ | $-11.12\%$ | $-7.15\%$ | $-5.49\%$ | $-5.58\%$ |
| 8 | ✗ | $40098 \pm 900$ | $21.17 \pm 0.45$ | $18682 \pm 580$ | $13593 \pm 1036$ | $19697 \pm 599$ |
| | ✓ | $37224 \pm 803$ | $18.86 \pm 1.21$ | $17104 \pm 579$ | $13486 \pm 735$ | $19355 \pm 555$ |
| diff(%) | | $-7.17\%$ | $-10.90\%$ | $-8.45\%$ | $-0.79\%$ | $-1.74\%$ |

sampling strategies: (1) fully recurrent behavior policy (no truncation), which causes confounding by conditioning on variables that are being truncated during training; (2) behavior policy with same truncation length as the target policy (see also Figure 6 for the causal graph). It is noteworthy that the confounded approach is actually widely used by popular algorithms (e.g., DRQN (Hausknecht and Stone, 2015)). We summarize the results in Table 2. Surprisingly, we find that this simple change leads to small, yet consistent performance degradation across all five environments and two truncation lengths. This suggests that the ALE may be more partially observable than previously believed, and confounding should be considered when designing sampling strategies.

In Appendix C.5, we also examine the effect of the latent space size $|\mathcal{Z}|$ on the performance, and find it to be relative robust above a certain level. This suggests that while the environments are stochastic, the stochasticity can be well approximated by a small number of latent variables.

## 5  RELATED WORK

**Advantage Estimation** Estimating the advantage function is an important part of policy optimization (Kakade and Langford, 2002). Schulman et al. (2015) proposed Generalized Advantage Estimation (GAE), which utilizes TD($\lambda$) (Sutton, 1988) to perform on-policy multi-step estimates of the advantage function. Wang et al. (2016) proposed dueling network to parametrize $Q_\theta$ into $V_\theta + A_\theta$ and showed that it can improve the performance of the original DQN. Tang et al. (2023) proposed VA learning to estimate $V$ and $A$ separately, and showed that it can outperform the dueling architecture. Pan et al. (2022) proposed DAE to perform on-policy multi-step estimation of the advantage function. This was later generalized to the off-policy setting by Pan and Schölkopf (2024). The present work extends off-policy DAE to partially observable environments and improves its computational efficiency.

**POMDP** POMDPs provide a general framework for studying decision making with incomplete states (Åström, 1965). In RL, POMDPs are usually solved by first converting them into MDPs either using belief states (Kaelbling et al., 1998) or information vectors (Bertsekas, 2012). In deep RL, partial observability is usually addressed using frame-stacking (Mnih et al., 2015), or by modeling the histories directly (Kapturowski et al., 2018; Gruslys et al., 2018; Hafner et al., 2023; Hausknecht and Stone, 2015; Mnih et al., 2016).

**Latent Dynamics Model** Learning dynamics models in the latent space is a promising approach to model-based RL (Ha and Schmidhuber, 2018; Han et al., 2019; Schrittwieser et al., 2020; Hafner et al., 2023; Antonoglou et al., 2021). It is, however, still common to rely on reconstructing observations to learn meaningful latent representations (Anand et al., 2021). In the present work, we combine ideas from self-supervised learning methods (Schwarzer et al., 2020; Grill et al., 2020) and the WTA loss (Makansi et al., 2019; Rupprecht et al., 2017) to estimate the transition probabilities purely in the latent space, and found it to be beneficial.

**Causality** The problem of inferring the effect of an action under partial observability dates at least back to Splawa-Neyman et al. (1990); Rubin (1974), and is a central topic in the study of causal inference (Pearl, 2009; Peters et al., 2017). In RL, these problems have been studied in the bandit setting (Bareinboim et al., 2015; Tennenholtz et al., 2021) and the sequential setting (Tennenholtz

et al., 2020; Pace et al., 2023). We showed that the confounding problem can also have negative impacts when training with recurrent policies.

## 6 DISCUSSION

In the present work, we showed how to extend DAE for POMDPs and addressed the computational cost issue by using discrete latent dynamics models. Through experiments in the ALE, we demonstrated that DAE is sample efficient and scalable, and that the proposed corrections are effective.

One limitation of our method is the need to approximate the transition probabilities through the use of latent dynamics. This introduces additional hyperparameters (e.g., network architectures of the dynamics model), and renders our method closer to model-based than model-free, although we do not explicitly use the model for rollouts. One direction for future work is to explore model-free approaches to approximate the constraints. Another limitation is that, while we can partially mitigate the problem of confounding caused by using truncated trajectories, our approach is only applicable in the online setting, where we can control the behavior policy. An important direction is to develop computationally efficient methods for eliminating the confounding bias for broader settings.

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

# A    PROOF OF PROPOSITION 1

**Proposition** (Off-policy DAE for POMDPs). *Given behavior policy $\mu$, target policy $\pi$, and backup length $n \geq 0$. $(A^\pi, B^\pi, V^\pi)$ is a minimizer of*

$$
\mathcal{L}(\hat{A}, \hat{B}, \hat{V}) = \mathbb{E}_\mu \left[ \left( \sum_{t'=0}^{n-1} \gamma^{t'} \left( r_{t+t'} - \hat{A}_{t+t'} - \hat{B}_{t+t'} \right) + \gamma^n \hat{V}(h_{n+t}) - \hat{V}(h_t) \right)^2 \right]
$$

$$
subject\ to\ \begin{cases} \mathbb{E}_{a \sim \pi(\cdot|h)}[\hat{A}(h,a)|h] = 0 & \forall h \in \mathcal{H} \\ \mathbb{E}_{(r,o') \sim p(\cdot|h,a)}[\hat{B}(h,a,r,o')|h,a] = 0 & \forall (h,a) \in \mathcal{H} \times \mathcal{A} \end{cases},
$$

(13)

*where $\mathcal{H}$ is the set of all trajectories of the form $(o_0, a_0, r_0, ...o_t)$, $\hat{A}_t = \hat{A}(h_t, a_t)$, and $\hat{B}_t = \hat{B}(h_t, a_t, r_t, o_{t+1})$. Furthermore, the minimizer is unique if, for any trajectory $h \in \mathcal{H}$, $p_\mu(h) > 0$.*

*Proof.* Firstly, we note that a POMDP can be reformulated as an MDP with state space equal to the space of information vectors ($h_t$) (Bertsekas, 2012). The theorem is then a direct result of applying Off-policy DAE (Pan and Schölkopf, 2024) to the reformulated MDP.  □

Remark: The original proof of Off-policy DAE assumes that the reward function is deterministic, which can be violated when converting POMDPs into MDPs. As such, our definition of $B^\pi(s,a,r,s') = r + \gamma V^\pi(s') - \mathbb{E}_{\pi,s'' \sim p(\cdot|s,a)}[r + \gamma V^\pi(s'')|s,a]$ (in a fully observable MDP) differs slightly from the original one $B^\pi(s,a,s') = \gamma V^\pi(s') - \mathbb{E}_{\pi,s'' \sim p(\cdot|s,a)}[\gamma V^\pi(s'')|s,a]$.

# B    CAUSAL GRAPH OF TRUNCATED SEQUENCES

Figure 6 shows the causal relationship between variables when sequences are truncated. For multi-step methods like DAE, we learn the value/advantage functions by building a model that takes in $(h'', a_t, r_t, o_{t+1}, \cdots)$ to predict $\sum_{t'>t} r_{t'}$ (assuming the backup length is infinity for illustrative purpose). It is then clear that $h'$ can influence both the input variables and the output variables, and lead to confounding. In the confounding experiment in section 4, the two sampling strategies differ in whether the red arrows are present for the behavior policy.

Modern recurrent agents (e.g., R2D2 (Kapturowski et al., 2018), Dreamer (Hafner et al., 2023)) often store recurrent states in the replay buffer during sampling and initialize the RNN states from the replay buffer during training. While, in theory, this can mitigate the confounding bias if the stored recurrent states contain enough information to predict future observables (i.e., the recurrent states are sufficient statistics), these methods are harder to analyze, as it is difficult to quantify (or measure) the quality of the recurrent states. Consequently, we only consider the simplest case (no recurrent states), and leave it for future work to explore other directions.

# C    EXPERIMENT DETAILS & ADDITIONAL RESULTS

## C.1    PSEUDOCODE AND ADDITIONAL IMPLEMENTATION DETAILS

We provide the pseudocode in Algorithm 1. For illustrative purpose, the pseudocode assumes a single actor and batch size 1; however, the algorithm can be easily parallelized over multiple actors and mini-batches.

To avoid the latent dynamics from collapsing, we use a soft loss for the reconstruction by including $\epsilon_W \geq 0$ into the posterior construction. In practice, $\epsilon_W$ is linearly annealed from 1 to 0 in the early stage of training. This is similar to the approach proposed by Makansi et al. (2019), which was found to make training less dependent on initialization, except that the authors construct the posterior using the top-$k$ nearest neighbors.

Incorporating stochastic rewards can be done by adding an additional reward reconstruction loss. In the case of Atari games, we can exploit the discrete structure of the rewards (rewards can only be in $\mathcal{R} = \{-1, 0, 1\}$) and construct the latent space by $\mathcal{Z} = \mathcal{Z}_O \times \mathcal{R}$. This then allows us to

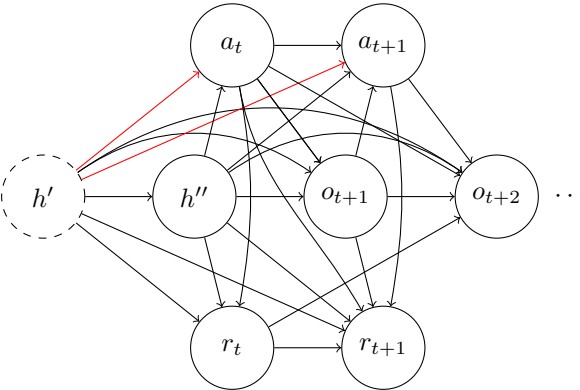

Figure 6: Causal relationship between variables of a truncated sequence for a general POMDP. $h' = h_{0:t-k-1}$ denotes the truncated part of the sequence, and $h'' = h_{t-k:t}$ denotes the remaining (or "context") part of the sequence. The red arrows shows the dependency between actions and $h'$ when using recurrent actors.

decompose the prior and the posterior by $p(z|h,a) = p(z_o|h,a)p(r|h,a)$ and $p(z|h,a,r,o') = p(z_o|h,a,r,o')p(\hat{r}|h,a,r,o')$, respectively. Note that $p(\hat{r}|h,a,r,o') = \mathbb{I}(\hat{r} = r)$ is simply the indicator function.

As pointed out by Pan et al. (2022), having a smoothly changing target policy is crucial to optimizing the DAE objective function. Consequently, we use a softmax policy based on $\hat{A}_{\theta_{\text{EMA}}}$ as the target policy. However, as reward densities can vary drastically between environments, we additionally learn a temperature parameter $T$ by minimizing $\log T + \beta_{\text{KL}} \text{KL}(\pi || \pi_{\text{EMA}})$, where both policies are softmax policies constructed using the advantage functions (i.e. $\pi = \text{softmax}(\frac{\hat{A}}{T})$). This ensures that the online policy $\pi$ does not deviate too much from the target policy $\pi_{\text{EMA}}$, and alleviates the need to tune the temperature manually for each environment.

Finally, to balance the scales between various objective functions, we set $\beta_V$ to be inverse proportional to the standard deviation of the cumulative rewards (i.e., $\sigma(G)$).

### C.2 ENVIRONMENT SETTING

For fair comparison, our environment settings follow the ones used by the Dopamine baseline (Castro et al., 2018), except that we do not use frame-stacking. In addition, we use EnvPool (Weng et al., 2022) for efficient implementation of the parallelized environments.

| Parameter | Value |
|---|---|
| Grey-scaling | True |
| Observation Resolution | 84×84 |
| Frame Stack | 1 |
| Action Repetitions | 4 |
| Reward Clipping | [-1, 1] |
| Terminal on life-loss | False |
| Sticky Action Prob. | 0.25 |
| $\gamma$ (discount factor) | 0.99 |

Table 3: ALE preprocessing parameters. Blue: Best practice suggested by Machado et al. (2018). Red: Differ from the baseline (Castro et al., 2018).

### C.3 HYPERPARAMETERS

Table 4 summarizes the default hyperparameters used in the experiments. The hyperparameters largely follows the ones used by Castro et al. (2018) with some exceptions. For the learning rate, we

**Algorithm 1** Off-policy DAE (POMDP)

**Require:** $n$ (backup length), $k$ (truncation length)
1: Initialize network parameters $\theta$
2: $\theta_{\text{EMA}} \leftarrow \theta$
3: $D = \{\}$
4: Observe $o_0$
5: $h_0 \leftarrow (o_0)$
6: **for** $t = 0, 1, 2, \ldots$ **do**
7:    Sample transition $(o, a, r, o')$ with $\epsilon$-greedy based on $\hat{A}_\theta(h_t, \cdot)$
8:    $h_{t+1} \leftarrow (h_t, a, r, o')$
9:    $h_{t+1} \leftarrow h_{t+1-k:t+1}$ (truncation)
10:    $D \leftarrow D \cup \{(o, a, r, o')\}$
11:    **if** $t + 1 \bmod \texttt{steps\_per\_update} = 0$ **then**
12:       Sample an $n + k$-step trajectory $\mathcal{T} = (o_i, a_i, r_i, \ldots, o_{i+n+k})$ from $D$
13:       Encode observations of $o_i$ into $x_i$
14:       Compute the predicted next embedding $\hat{x}_{i+1}$ for each time step $i$
15:       Compute the posterior

$$p(z|h_i, a_i, x_{i+1}) = \begin{cases} 1 - \epsilon_{\text{WTA}} + \frac{\epsilon_{\text{WTA}}}{|\mathcal{Z}|}, & \text{if } z = \arg\min_z \|\hat{x}_{i+1,z} - x_{i+1}\| \\ \frac{\epsilon_{\text{WTA}}}{|\mathcal{Z}|}, & \text{otherwise} \end{cases}$$

16:       Compute embedding reconstruction loss by

$$\mathcal{L}_{\text{rec}} = \sum_{i>k} \sum_z p(z|h_i, a_i, x_{i+1}) \|\hat{x}_{i+1,z} - \texttt{sg}(x_{i+1})\|^2$$

17:       Compute prior loss $\mathcal{L}_{\text{prior}} = -\sum_{i>k} \log p_\theta(z_i|h_i, a_i)$
18:       Approximate $B$-constraint by

$$\hat{B}_{\theta,i} \leftarrow \sum_z \left( p(z|h_i, a_i, x_{i+1}) - \texttt{sg}(p_\theta(z|h_i, a_i)) \right) \hat{B}_\theta(h_i, a_i, z)$$

19:       Compute target policy $\pi_{\text{target}} \leftarrow \texttt{softmax}(\frac{\hat{A}_{\theta_{\text{EMA}}}}{T})$
20:       Compute online policy $\pi \leftarrow \texttt{softmax}(\frac{\hat{A}_\theta}{T})$
21:       Enforce $A$-constraint by

$$\hat{A}_{\theta,i} \leftarrow \hat{A}_\theta(h_i, a_i) - \sum_a \hat{A}_\theta(h_i, a) \pi_{\text{target}}(h_i, a)$$

22:       Compute DAE objective by (note that we truncate the first $k$ elements)

$$\mathcal{L}_{\text{DAE}} = \left( \sum_{j=k}^n \gamma^{j-k} (r_{i+j} - \hat{A}_{\theta,i+j} - \hat{B}_{\theta,i+j}) + \gamma^{n-k+1} \hat{V}_{\theta_{\text{EMA}}, i+n+k} - \hat{V}_{\theta,i} \right)^2$$

23:       Compute adaptive temperature objective $\mathcal{L}_T = \log T + \beta_{\text{KL}} \text{KL}(\pi \| \pi_{\text{target}})$
24:       Update $\theta$ by SGD with loss function $\beta_V \mathcal{L}_{\text{DAE}} + \beta_{\text{prior}} \mathcal{L}_{\text{prior}} + \beta_{\text{rec}} \mathcal{L}_{\text{rec}} + \mathcal{L}_T$
25:       $\theta_{\text{EMA}} \leftarrow \tau \theta_{\text{EMA}} + (1 - \tau)\theta$
26:    **end if**
27: **end for**

found linear warmup to be important, which is likely due to the use of LSTMs that can be unstable in the early stage of training. The batch size indicates the number of trajectories instead of frames, as such, the number of frames per batch is (backup length + truncation length) × batch size.

| Parameter | Value |
|---|---|
| Replay buffer size | 1000000 |
| Minimum Steps before training | 20000 |
| Number of parallel actors | 16 |
| $\epsilon$ (exploration) | Linearly annealed from 1 to 0.01 in the first 1M steps |
| $\epsilon$ (evaluation) | 0.001 |
| Optimizer | Adam (Kingma and Ba, 2014) |
| Learning rate | Linear warmup from 0 to $1.25 \times 10^{-4}$ in the first 100000 steps and then linearly annealed to 0 throughout training |
| Adam $\beta$ | (0.9, 0.95) |
| Adam $\epsilon$ | $10^{-6}$ |
| Replay ratio ($\frac{\text{Gradient updates}}{\text{Environment steps}}$) | 0.0625 |
| Backup length | 16 |
| Truncation length | 8 |
| Batch size | 12 |
| $|\mathcal{Z}|$ | 16 |
| $\epsilon_{\text{WTA}}$ | Linearly annealed from 1 to 0 in the first 500000 steps |
| $\tau$ (target EMA) | 0.995 |
| $\beta_{\text{prior}}$ | 0.025 |
| $\beta_{\text{rec}}$ | 1 |
| $\beta_{\text{KL}}$ | 150 |

Table 4: Default hyperparameters for the experiments.

## C.4 NETWORK ARCHITECTURE

Figure 7 shows the network architecture used in the experiments. In the scaling experiments, we only multiply the width of the convolutional layers in the ResNet by the multiplier, with the sizes of other modules fixed. Table 5 summarizes the number of parameters in each component.

We use Layer Normalization (Ba et al., 2016) before the activations in the MLP heads and before the LSTM. In addition, we apply L2 normalization to the image embeddings (after the linear layer) such that the SPR objective (cosine similarity) reduces to L2 distance between the encoded vectors.

In the ablation study, we replace the LSTM layer with a 1D convolution with kernel size 4 to simulate the effect of stacking 4 frames. This has the same effect of limiting the context window to 4, and can also be seen as a late-fusion type of network for video processing, in contrast to frame-stacking, which can be seen as early-fusion.

| Component | Parameters (millions) |
|---|---|
| ResNet | $2 \times m$ |
| LSTM | 7 |
| Transition Model (VQ-VAE) | 21 |
| Value heads ($\hat{A}, \hat{B}, \hat{V}$) | 3 |

Table 5: Number of parameters in each component.

## C.5 ADDITIONAL RESULTS

**Latent space size** The latent dynamics model relies on having multiple predictions to capture the stochasticity of the environment. Here we examine the impact of the number of predictions at each timestep on the learning performance. We summarize the results in Figure 10 and Table 6. In general, we find the agent's performance to be quite robust.

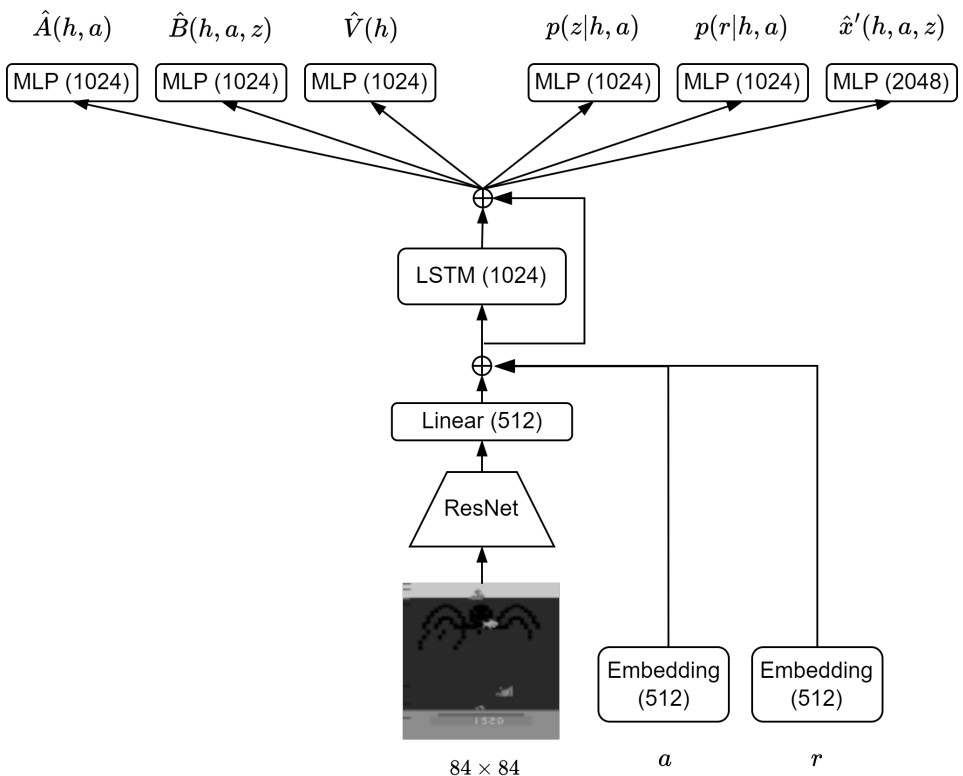

Figure 7: The network architecture. We use the same ResNet encoder proposed by Espeholt et al. (2018). All MLP heads have 1 hidden layer. Previous actions and rewards are first embedded into 512-dimensional vectors before summed together with the image embedding to form the final embedding vector. We use a residual connection around the LSTM similar to Kim et al. (2017).

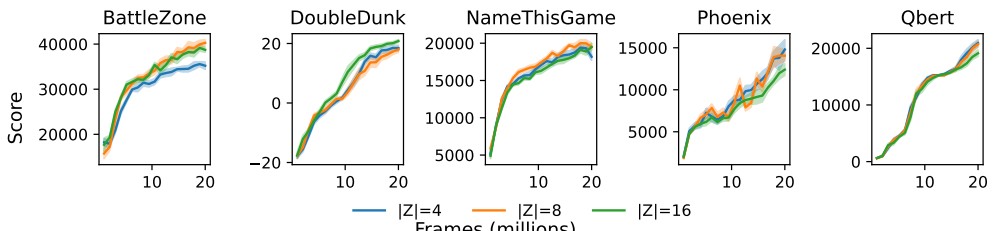

Figure 8: Effect of $|\mathcal{Z}|$ on the sample efficiency. Results are aggregated over 10 random seeds. Lines and shadings represent the mean and 1 standard error, respectively.

Table 6: Effect of latent space size on the final evaluation score. Scores were aggregated over 10 random seeds after 20M training frames. Values represent (mean)±(1 standard error).

| $|\mathcal{Z}|$ | BattleZone | DoubleDunk | NameThisGame | Phoenix | Qbert |
|---|---|---|---|---|---|
| 4 | $35738 \pm 584$ | $19.14 \pm 0.81$ | $18579 \pm 659$ | $15902 \pm 1065$ | $21161 \pm 695$ |
| 8 | $40044 \pm 1152$ | $18.50 \pm 0.93$ | $19805 \pm 466$ | $16163 \pm 1496$ | $20686 \pm 618$ |
| 16 | $40098 \pm 900$ | $21.17 \pm 0.45$ | $18682 \pm 580$ | $13593 \pm 1036$ | $19697 \pm 599$ |

**Exponential moving average**   While soft targets (EMA) were used in the original implementation of SPR (Schwarzer et al., 2020), hard targets (periodic copy) are much more popular among DQN variants. Here, we compare the effect of using soft and hard moving targets on the performance. For hard updates, we follow Rainbow (Hessel et al., 2018) and set the update period to 8000 agent steps (32k frames). Figure 9 and Table 7 summarize the effects of updating rules on learning curves and final performance, respectively. In general, we find EMA updates to be more effective.

Table 7: Effect of target update rules on the final evaluation score. Scores were aggregated over 10 random seeds after 20M training frames. Values represent (mean)±(1 standard error).

| Update | BattleZone | DoubleDunk | NameThisGame | Phoenix | Qbert |
|---|---|---|---|---|---|
| Soft | $40098 \pm 900$ | $21.17 \pm 0.45$ | $18682 \pm 580$ | $13593 \pm 1036$ | $19697 \pm 599$ |
| Hard | $36218 \pm 979$ | $15.34 \pm 1.47$ | $20513 \pm 427$ | $11536 \pm 259$ | $17890 \pm 682$ |

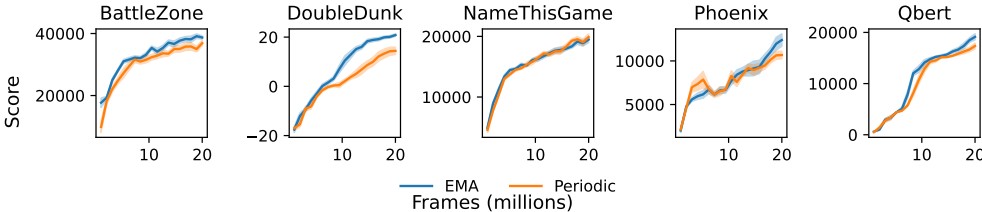

Figure 9: Effect of target updating rules on the sample efficiency. Results are aggregated over 10 random seeds. Lines and shadings represent the mean and 1 standard error, respectively.

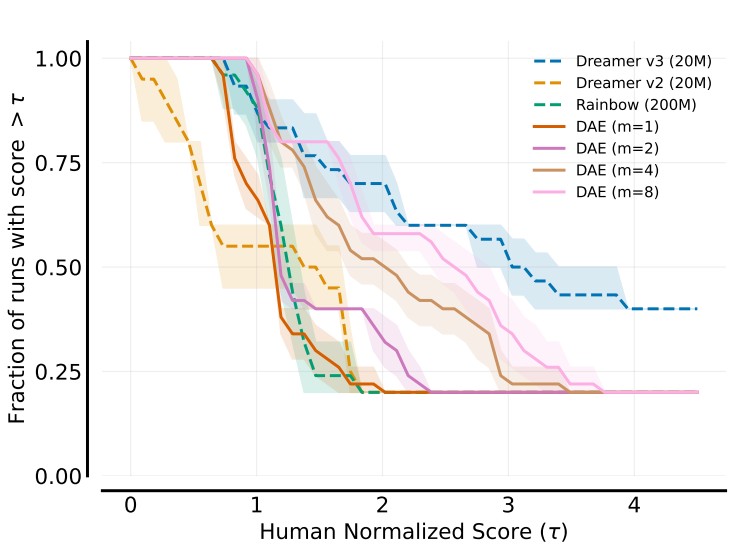

Figure 10: Performance profile (Agarwal et al., 2021) for Atari-5.

