# OpenReview forum: "Direct Advantage Estimation in Partially Observable Environments"
_ICLR.cc/2025/Conference — Submitted to ICLR 2025_

### Official Review · Reviewer_jrnv · 2024-10-24

**Soundness:** 3
**Presentation:** 3
**Contribution:** 2
**Rating:** 5
**Confidence:** 4

**Summary:**

This paper extend the off-policy Direct Advantage Estimation (DAE) technique into the POMDP environments with model estimation trick to reduce the complexity.

**Strengths:**

The idea is sound and motivation is intuitive with straightforward explanation, building upon the previous works of DAE and off-policy DAE[4,5]. The method itself has been proved effective on the Arcade Learning benchmarks. Authors have also conducted extensive experiments to validate hyper-parameters selection and various corrections.

**Weaknesses:**

1, line 247, 255, and 266 would be better to refer to Eq 9 instead of Eq 14 back in the appendix.\
2, Though technical sound, this paper seems a bit incremental comparing with previous works[4] with a different dynamic reparameterization for high-dim state space.\
3, Section 3.2 states the reduction of computational complexity comparing to LSTM and RNN based methods, but it would be great to have some empirical or theoretical demonstrations.\
4, The authors claim that off-policy correction is crucial for stochastic environment and performance improvement. It would be better to explain how the stochastic part is validated. \
5, Reviewer is also interested in seeing the comparison with other recent works with support for pomdp such as [1,2,3]. Current comparisons with Rainbow and DQN might not be sufficient.

[1] Hansen, Nicklas, Hao Su, and Xiaolong Wang. "Td-mpc2: Scalable, robust world models for continuous control." arXiv preprint arXiv:2310.16828 (2023).\
[2] Chen, Xiaoyu, et al. "Flow-based recurrent belief state learning for pomdps." International Conference on Machine Learning. PMLR, 2022.\
[3] Danijar Hafner, Timothy Lillicrap, Ian Fischer, Ruben Villegas, David Ha, Honglak Lee, and James Davidson. Learning latent dynamics for planning from pixels. In International conference on machine learning, pages 2555–2565. PMLR, 2019.\
[4]Pan, Hsiao-Ru, and Bernhard Schölkopf. "Skill or luck? return decomposition via advantage functions." arXiv preprint arXiv:2402.12874 (2024).\
[5]Pan, Hsiao-Ru, et al. "Direct advantage estimation." Advances in Neural Information Processing Systems 35 (2022): 11869-11880.

**Questions:**

1, Is it necessary to incorporate reward signal $r_{0:t-1}$ to the information vector $h_t$? Since $r_t$ info seems not be incorporated in the $h_t$.\
2, The reviewer is a little bit confusing on the mixing use of state $s$, $h$, $z$, and $x$. It would be great to have a better clarification on these symbols and definitions of corresponding $\hat{A}, \hat{B},$ and $\hat{V}$.\

---

> ### Author Response · Authors · 2024-11-19
> **Official Comment by Authors (1/2)**
>
> We thank the reviewer for the detailed and constructive feedback, please see our answers to your questions below.
>
> > line 247, 255, and 266 would be better to refer to Eq 9 instead of Eq 14 back in the appendix.
>
> We fixed this in the latest revision.
>
> > Though technical sound, this paper seems a bit incremental comparing with previous works[4] with a different dynamic reparameterization for high-dim state space.
>
> We would like to emphasize that, this seemingly incremental change to the dynamics model can have far-reaching consequences. Previously, Off-policy DAE’s empirical results were limited to environments with low-dimensional states, precisely because learning a stochastic dynamics model was too computationally expensive. As a result, it remained unclear whether Off-policy DAE would also be effective in high-dimensional domains. The approach presented in this work –combining SPR, a representation learning method, with the WTA loss, a method for multi-modal predictions, to approximate the dynamics–represents, to the best of our knowledge, a novel approach to this problem that can be easily applied to high-dimensional domains. Furthermore, our empirical results not only validated the effectiveness of this approach, but also demonstrated the viability of Off-policy DAE in high-dimensional domains, which, we believe, is a valuable contribution.
>
> > Section 3.2 states the reduction of computational complexity comparing to LSTM and RNN based methods, but it would be great to have some empirical or theoretical demonstrations.
>
> The method of truncating sequences, presented in Section 3.2, is not intended to replace RNN-based methods, but rather to reduce their computational complexity. This reduction in computational complexity is straightforward, as parts of the trajectories are discarded when forwarding them through an RNN. While this method is widely adapted in deep RL, Section 3.2 highlights a commonly overlooked problem (i.e., confounding) that can happen when applying this method. We have revised Section 3.2 to better explain how this method is typically implemented in practice and its connection to confounding.
>
> > The authors claim that off-policy correction is crucial for stochastic environment and performance improvement. It would be better to explain how the stochastic part is validated.
>
> We have expanded the explanation regarding the off-policy correction in the latest revision. Essentially, the idea is that, for deterministic environments, we have $B^\pi\equiv 0$ for arbitrary policy $\pi$, and this term only comes into play when the environment is stochastic. As such, we can view the $\hat{B}$ terms in the DAE objective function as corrections for stochastic dynamics, and by forcing $\hat{B}\equiv 0$, we can see the importance of off-policy corrections in stochastic environments. This trick was also used in [1] to demonstrate the effectiveness of the corrections in stochastic environments.
>
> > Reviewer is also interested in seeing the comparison with other recent works with support for pomdp such as [1,2,3]. Current comparisons with Rainbow and DQN might not be sufficient.
>
> As TD-MPC2 and FORBES were originally developed for continuous action domains, it remains unclear whether they would also work well for discrete action domains like Atari games. As a remedy, we have updated our list to include more recent SOTA model-based recurrent algorithms DreamerV2 [2] and Dreamer V3 [3] (evaluated at 20M training frames, same training frames as ours). We summarize the results in the following table:
>
> | Method   |  BattleZone | DoubleDunk | NameThisGame | Phoenix | Qbert |
> |-|-|-|-|-|-|
> | Our (m=1) |  $35044\pm 986$ | $8.80\pm 2.13$ | $10977\pm 441$ | $5666\pm 54$ | $15313\pm 56$ |
> | Our (m=2) |  $38164\pm 603$ | $17.68\pm 1.15$ | $14308\pm 289$ | $8010\pm 270$ | $15831\pm 146$ |
> | Our (m=4) |  $40098\pm 900$ | $21.17\pm 0.45$ | $18682\pm 580$ | $13593\pm 1036$ | $19697\pm 599$ |
> | Our (m=8) | $39262\pm 763$ | $21.49\pm 0.38$ | $19638\pm 633$ | $18127\pm 1075$ | $23451\pm 415$ |
> | Rainbow | $40061\pm 1866$ | $22.12\pm 0.34$ | $9026\pm 193$ | $8545\pm 1286$ | $17383\pm 543$ |
> | DreamerV2 | $21225\pm743$ | $12.95\pm1.31$ | $12145\pm87$ | $9117\pm2151$ | $3441\pm969$ |
> | DreamerV3 | $37818\pm3979$ | $22.46\pm0.35$ | $20652\pm1002$ | $56889\pm3402$ | $23540\pm1976$ |
>
>
>  We note that DreamerV3 uses a significantly larger model (\~200M parameters) compared to our m=8 model (\~50M parameters). To enable a closer comparison, we also include Dreamer V2 (\~20M parameters). The results show that our method (m=8) achieves performance close to DreamerV3 in 3 out of 5 environments, while our smaller models (m=2, \~35M parameters) outperforms DreamerV2 in 4 out of 5 environments. Finally, while we also learn dynamics models for our methods, we differ from Dreamer in that we do not use these models for rollouts. Instead, we leverage them solely for the off-policy corrections in the DAE objective (the B constraint).

---

> ### Author Response · Authors · 2024-11-19
> **Official Comment by Authors (2/2)**
>
> > Is it necessary to incorporate reward signal $r_{0:t−1}$ to the information vector $h_t$? Since $r_t$ info seems not be incorporated in the $h_t$.
>
> Whether reward signals should be incorporated into the information vector depends on how the reward function is defined. Intuitively, if the reward function depends on the state (i.e., $r: \mathcal{S} \times \mathcal{A} \rightarrow \mathbb{R}$, the setting used in the present work) [4], then rewards should be incorporated into the information vector as they might contain additional information about the state. On the other hand, if the reward function is defined as $r: \Omega \times \mathcal{A} \rightarrow \mathbb{R}$ [5], then rewards are not needed as past observations/actions already form a sufficient statistic [6]. $r_t$ is not incorporated into $h_t$ because, under our notation, $r_t$ is observed after taking action $a_t$.
>
> > The reviewer is a little bit confusing on the mixing use of state $s, h, z,$ and $x$. It would be great to have a better clarification on these symbols and definitions of corresponding $\hat{A},\hat{B}$, and $\hat{V}$.
>
> We have updated Section 3 to make the notations more consistent, and Figure 1 to better illustrate the relationships between variables.
>
> [1] Pan, Hsiao-Ru, and Bernhard Schölkopf. "Skill or luck? return decomposition via advantage functions." arXiv preprint arXiv:2402.12874 (2024).
>
> [2] Hafner, Danijar, et al. "Mastering atari with discrete world models." arXiv preprint arXiv:2010.02193 (2020).
>
> [3] Hafner, Danijar, et al. "Mastering diverse domains through world models." arXiv preprint arXiv:2301.04104 (2023).
>
> [4] Kaelbling, Leslie Pack, Michael L. Littman, and Anthony R. Cassandra. "Planning and acting in partially observable stochastic domains." Artificial intelligence 101.1-2 (1998): 99-134.
>
> [5] Liu, Qinghua, et al. "When is partially observable reinforcement learning not scary?." Conference on Learning Theory. PMLR, 2022.
>
> [6] Bertsekas, Dimitri P. "Dynamic programming and optimal control 3rd edition, volume ii." Belmont, MA: Athena Scientific 1 (2011).

---

> > ### Comment · Reviewer_jrnv · 2024-11-22
> >
> > Reviewer jrnv thanks for authors' effort to improve the paper.  Some answers do solve reviewer's questions, but here are some remaining concerns.
> >
> > * Could author highlight the changing part in the updated version? It is little bit hard to spot those changes, particularly in section 3 and experiment part.
> > * Dreamer-v2, Dreamer-v3, and Rainbow are trained under 200M steps. Since authors' statement is not about sample efficiency advantage over Dreamer-v2 and Dreamer-v3, reviewer think the result might not be convincing under 20M training steps, since both methods training are not converged yet.
> > * Since paper is concentrating on discrete action domain, the comparison on only 5 atari games might not be convincing enough to claim advantages. Not saying on all 60 atari games shown in [1,2,3], but definitely more are needed. Maybe authors can also consider adding experiments in different test-bench such as ProcGen[4].
> > * Since proposed method does not have a dominated results comparing to Dreamer-v3[2], reviewer think comparison with MuZero[3] is needed.
> >
> > [1]Hafner, Danijar, et al. "Mastering atari with discrete world models." arXiv preprint arXiv:2010.02193 (2020).
> > [2]Hafner, Danijar, et al. "Mastering diverse domains through world models." arXiv preprint arXiv:2301.04104 (2023).
> > [3]Schrittwieser, Julian, et al. "Mastering atari, go, chess and shogi by planning with a learned model." Nature 588.7839 (2020): 604-609.
> > [4]Cobbe, Karl, et al. "Leveraging procedural generation to benchmark reinforcement learning." International conference on machine learning. PMLR, 2020.

---

> > > ### Author Response · Authors · 2024-11-22
> > >
> > > We thank the reviewer for the efforts, which truly helped us improve our paper. Please see our response below:
> > >
> > > > Could author highlight the changing part in the updated version? It is little bit hard to spot those changes, particularly in section 3 and experiment part.
> > >
> > > We have uploaded a revision which highlights the major changes in blue. In summary, we revised section 3.1 to avoid mixing usages of the state variables $s$ and the information vector $h$. The updated Figure 1 now also illustrates the relationships between $o$, $x$ and $h$ more clearly now.
> > >
> > > > Dreamer-v2, Dreamer-v3, and Rainbow are trained under 200M steps. Since authors' statement is not about sample efficiency advantage over Dreamer-v2 and Dreamer-v3, reviewer think the result might not be convincing under 20M training steps, since both methods training are not converged yet.
> > >
> > > We thank the reviewer for pointing this out. We have revised the statement to emphasize that our method is comparable to Dreamers in terms of sample efficiency up to 20M frames. Additionally, we would also like to point out that the Dreamer methods use the learned model to generate synthetic trajectories, effectively increasing the number of samples seen by the actor/critic, whereas our method only uses the model to enforce the constraint.
> > >
> > > > Since paper is concentrating on discrete action domain, the comparison on only 5 atari games might not be convincing enough to claim advantages. Not saying on all 60 atari games shown in [1,2,3], but definitely more are needed. Maybe authors can also consider adding experiments in different test-bench such as ProcGen[4].
> > >
> > > We would like to emphasize that the 5 games set were not chosen at random, but picked specifically based on [1], which demonstrated that these 5 games have very strong predictive power of the overall performance of an algorithm.
> > >
> > > While ProcGen also uses discrete actions, the main challenge of ProcGen (generalization to unseen levels), is very different from the main focus of this paper (partial observability), and the problem of partial observability in Atari games has been demonstrated by [2].
> > >
> > > > Since proposed method does not have a dominated results comparing to Dreamer-v3[2], reviewer think comparison with MuZero[3] is needed.
> > >
> > > We compare our method to MuZero [3] (an updated version of the original MuZero) below. Since the authors only provided the averaged curve, we only report the averaged scores for MuZero:
> > >
> > > | Method   |  BattleZone | DoubleDunk | NameThisGame | Phoenix | Qbert |
> > > |-|-|-|-|-|-|
> > > | Our (m=1) |  $35044\pm 986$ | $8.80\pm 2.13$ | $10977\pm 441$ | $5666\pm 54$ | $15313\pm 56$ |
> > > | Our (m=2) |  $38164\pm 603$ | $17.68\pm 1.15$ | $14308\pm 289$ | $8010\pm 270$ | $15831\pm 146$ |
> > > | Our (m=4) |  $40098\pm 900$ | $21.17\pm 0.45$ | $18682\pm 580$ | $13593\pm 1036$ | $19697\pm 599$ |
> > > | Our (m=8) | $39262\pm 763$ | $21.49\pm 0.38$ | $19638\pm 633$ | $18127\pm 1075$ | $23451\pm 415$ |
> > > | MuZero (20M frames) | $19971$ | $-17.67$ | $3084$ | $2341$ | $7040$ |
> > > | MuZero (40M frames) | $35787$ | $-9.93$ | $11406$ | $6503$ | $17645$ |
> > > | MuZero (60M frames) | $41873$ | $4.43$ | $24246$ | $19464$ | $22555$ |
> > >
> > > In general, we find that our smallest model is already competitive with MuZero under the 20M frames budget, and is much more efficient when scaled up.
> > >
> > > We thank the reviewer again and hope this addresses your concerns. Please let us know if you have any further questions.
> > >
> > > [1] Aitchison, Matthew, Penny Sweetser, and Marcus Hutter. "Atari-5: Distilling the arcade learning environment down to five games." International Conference on Machine Learning. PMLR, 2023.
> > >
> > > [2] Kapturowski, Steven, et al. "Recurrent experience replay in distributed reinforcement learning." International conference on learning representations. 2018.
> > >
> > > [3] Schrittwieser, Julian, et al. "Online and offline reinforcement learning by planning with a learned model." Advances in Neural Information Processing Systems 34 (2021): 27580-27591.

---

### Official Review · Reviewer_Pvss · 2024-10-31

**Soundness:** 2
**Presentation:** 1
**Contribution:** 2
**Rating:** 5
**Confidence:** 2

**Summary:**

This paper presents an extension of the off-policy Direct Advantage Estimation (DAE) method to partially observable Markov Decision Processes (POMDPs). The proposed approach demonstrates improved performance on certain benchmark problems compared to DQN. However, the notation throughout the paper is unclear, making it difficult to follow the methodology. Additionally, the baseline algorithm used for comparison is outdated, limiting the persuasiveness of the results. The motivation for applying DAE in this context is also insufficiently explained, and I find the necessity of the proposed method unconvincing.

**Strengths:**

1. This paper extend off-policy DAE to POMDP with minor modifications.
2. Authors address the problem of increasing computational costs by modeling the latent space transition with RNN.
3. Partial experimental results demonstrate that the proposed method has good performances.

**Weaknesses:**

1. The paper is hard for me to follow, and the writing could benefit from further polishing. For instance, the parameter notation $\theta$ is used in various forms, such as $f_\theta(h, a)$, $g_\theta(s, a, z)$, and $p_\theta(z|h_t, a_t)$. This raises questions about whether these functions share the same parameters, which the paper does not clarify. Additionally, the notation for $z$ in Eq. (10) is undefined until $\mathcal{Z}$ is introduced before Eq. (11), making the notation confusing.

2. The authors appear to consider only the simplest case, with finite state and action spaces. Under these conditions, why is DAE necessary, and why wouldn’t a TD loss with a Bellman operator suffice? The motivation for introducing DAE in this context is unclear and could be better presented.

3. The benchmark algorithms used in this paper date back 6-9 years, which weakens the strength of the comparison. I recommend that the authors incorporate more recent, robust baselines to more convincingly showcase the advantages of their proposed method.

**Questions:**

Same as the weakness.

---

> ### Author Response · Authors · 2024-11-19
>
> We thank the reviewer for the detailed and constructive feedback, please see our answers to your questions below.
>
> > The paper is hard for me to follow, and the writing could benefit from further polishing. For instance, the parameter notation $θ$  is used in various forms, such as $f_θ(h,a), g_θ(s,a,z)$, and $p_θ(z|h_t,a_t)$. This raises questions about whether these functions share the same parameters, which the paper does not clarify. Additionally, the notation for $z$ in Eq. (10) is undefined until $\mathcal{Z}$ is introduced before Eq. (11), making the notation confusing.
>
> We have updated Section 3 to make the notations more consistent. In general, $f_θ$, $g_θ$ and $p_θ$ need not share the same parameters, although in practice we do share parts of the parameters (the CNN and the RNN backbones) to make the forward pass more efficient. We have also updated the experiment section to reflect this design choice.
>
> > The authors appear to consider only the simplest case, with finite state and action spaces. Under these conditions, why is DAE necessary, and why wouldn’t a TD loss with a Bellman operator suffice? The motivation for introducing DAE in this context is unclear and could be better presented.
>
> While the convergence properties of TD algorithms are well-understood in tabular settings (finite state/action spaces), such analyses become much more challenging in the function approximation setting, especially within deep RL, due to the complexity of neural networks. As such, much of the research in deep RL has been empirical, and it remains unclear to what extent results from tabular settings can be transferred to this more complex setting. For instance, [1] demonstrates that convergent algorithms in tabular settings can sometimes diverge in deep RL settings.
>
> The motivation for introducing DAE, as outlined in the abstract (line 10) and the introduction (line 34-40), stems from previous works that have demonstrated the effectiveness of DAE in deep RL settings [2, 3]. We view the theoretical results in the tabular setting from the present work as initial steps toward justifying DAE’s application in the POMDP setting, with our experimental results providing empirical evidence that DAE can indeed perform effectively in the deep RL setting.
>
> > The benchmark algorithms used in this paper date back 6-9 years, which weakens the strength of the comparison. I recommend that the authors incorporate more recent, robust baselines to more convincingly showcase the advantages of their proposed method.
>
> We have updated our list to include more recent SOTA model-based recurrent algorithms Dreamer V2 [2] and Dreamer V3 [3] (evaluated at 20M training frames, same training frames as ours). We summarize the results in the following table:
>
> | Method   |  BattleZone | DoubleDunk | NameThisGame | Phoenix | Qbert |
> |-|-|-|-|-|-|
> | Our (m=1) |  $35044\pm 986$ | $8.80\pm 2.13$ | $10977\pm 441$ | $5666\pm 54$ | $15313\pm 56$ |
> | Our (m=2) |  $38164\pm 603$ | $17.68\pm 1.15$ | $14308\pm 289$ | $8010\pm 270$ | $15831\pm 146$ |
> | Our (m=4) |  $40098\pm 900$ | $21.17\pm 0.45$ | $18682\pm 580$ | $13593\pm 1036$ | $19697\pm 599$ |
> | Our (m=8) | $39262\pm 763$ | $21.49\pm 0.38$ | $19638\pm 633$ | $18127\pm 1075$ | $23451\pm 415$ |
> | Rainbow | $40061\pm 1866$ | $22.12\pm 0.34$ | $9026\pm 193$ | $8545\pm 1286$ | $17383\pm 543$ |
> | DreamerV2 | $21225\pm743$ | $12.95\pm1.31$ | $12145\pm87$ | $9117\pm2151$ | $3441\pm969$ |
> | DreamerV3 | $37818\pm3979$ | $22.46\pm0.35$ | $20652\pm1002$ | $56889\pm3402$ | $23540\pm1976$ |
>
> We note that DreamerV3 uses a significantly larger model (\~200M parameters) compared to our m=8 model (\~50M parameters). To enable a closer comparison, we also include DreamerV2 (\~20M parameters). The results show that our method (m=8) achieves performance close to DreamerV3 in 3 out of 5 environments, while our smaller models (m=2, \~35M parameters) outperforms DreamerV2 in 4 out of 5 environments. Finally, while we also learn dynamics models for our methods, we differ from Dreamer in that we do not use these models for rollouts. Instead, we leverage them solely for the off-policy corrections in the DAE objective (the B constraint).
>
> [1] Van Hasselt, Hado, et al. "Deep reinforcement learning and the deadly triad." arXiv preprint arXiv:1812.02648 (2018).
>
> [2] Hafner, Danijar, et al. "Mastering atari with discrete world models." arXiv preprint arXiv:2010.02193 (2020).
>
> [3] Hafner, Danijar, et al. "Mastering diverse domains through world models." arXiv preprint arXiv:2301.04104 (2023).

---

> > ### Comment · Reviewer_Pvss · 2024-11-26
> >
> > If the original intent of the DAE is that the Q and V functions learned and fitted through TD loss are inherently difficult to interpret, can the function fitting employed in DAE itself be clarified? I remain unconvinced by the motivation behind DAE, particularly its extension into POMDP.
> >
> > Taking into account the feedback from other reviewers, I will retain my score and level of confidence.

---

> > > ### Author Response · Authors · 2024-11-26
> > >
> > > Thank you for the comment
> > >
> > > > If the original intent of the DAE is that the Q and V functions learned and fitted through TD loss are inherently difficult to interpret, can the function fitting employed in DAE itself be clarified?
> > >
> > > We would like to clarify that we do not claim that the Q/V functions learned through TD loss are inherently more difficult to interpret than those learned by DAE. Nowhere in the paper do we make such an argument, nor do we consider it as a motivation for the present work.
> > >
> > > To reiterate, the main motivation, as stated in the introduction, is to extend the DAE framework, which has demonstrated strong empirical performance in the deep RL setting for fully observable environments, to POMDPs.
> > > Specifically, previous research has shown that using multi-step methods can accelerate and stabilize learning in deep RL [1, 2, 3], and DAE falls into this category, as it also leverages multi-step returns for its estimations.
> > > What sets DAE apart from other methods is its novel approach to off-policy corrections, which are often ignored in previous works, and the approach was shown to be effective in MDPs.
> > > Since the present work focuses on extending DAE to new domains rather than proposing DAE as a novel method, our motivation stems from its strong empirical performance and its limitations (such as partial observability and computational complexity), as highlighted in [4].
> > >
> > > We hope this addresses your concern, Please let us know if there are further questions.
> > >
> > > [1] Van Hasselt, Hado, et al. "Deep reinforcement learning and the deadly triad." arXiv preprint arXiv:1812.02648 (2018).
> > >
> > > [2] Hessel, Matteo, et al. "Rainbow: Combining improvements in deep reinforcement learning." Proceedings of the AAAI conference on artificial intelligence. Vol. 32. No. 1. 2018.
> > >
> > > [3] Hernandez-Garcia, J. Fernando, and Richard S. Sutton. "Understanding multi-step deep reinforcement learning: A systematic study of the DQN target." arXiv preprint arXiv:1901.07510 (2019).
> > >
> > > [4] Pan, Hsiao-Ru, and Bernhard Schölkopf. "Skill or luck? return decomposition via advantage functions." arXiv preprint arXiv:2402.12874 (2024).

---

### Official Review · Reviewer_6QRc · 2024-11-03

**Soundness:** 4
**Presentation:** 2
**Contribution:** 2
**Rating:** 6
**Confidence:** 3

**Summary:**

The paper extends and improves upon Off-Policy Direct Advantage Estimation, both by extending it to the POMDP case by modifying the way the loss is computed, but also by modelling the transitions in the latent space of a VQ-VAE, which improves the computational cost of training models with DAE.

**Strengths:**

The theoretical analysis is quite complete and sound, and the authors bring techniques from other subfields of Reinforcement Learning (World Models) to improve DAE and make it more practical.

**Weaknesses:**

The experimental analysis is rather weak, they compare their model with Rainbow and DQN, while using a recurrent and residual architecture signifcantly more complex and costly. Comparing with Impala and R2D2 would have been more meaningful. Besides that they report only mean and 1 standard error of their agents, whereas reporting the median agent performance and either a 2 standard deviation interval or better yet a tolerance interval would have improved the visualization of their algorithm performance.

**Questions:**

1. Would it be possible to compare the performance with methods using similar networks to the ones you used, like Impala and R2D2? Why did you think this comparison was irrelevant?
2. Would it be possible to report ablations on using a Exponential Moving Average network vs a periodically updated network (DQN-style) as your target network? If not would it be possible to justify the choice a bit more?

---

> ### Author Response · Authors · 2024-11-19
>
> We thank the reviewer for the detailed and constructive feedback, please see our answers to your questions below.
>
> > ... Besides that they report only mean and 1 standard error of their agents, whereas reporting the median agent performance and either a 2 standard deviation interval or better yet a tolerance interval would have improved the visualization of their algorithm performance.
>
> Currently, we reported performance measures for individual environments instead of the aggregated median performance because we only focus on a small subset of the full Atari suite. This allows us to more closely examine the impact of various corrections on specific environments (e.g., some environments are more partially observable, where LSTMs are more effective). As a remedy, we have included an additional performance profile as suggested by [1] in Figure 10. This profile displays the distribution of human-normalized scores along with bootstrapped confidence intervals.
>
> > Would it be possible to compare the performance with methods using similar networks to the ones you used, like Impala and R2D2? Why did you think this comparison was irrelevant?
>
> We certainly do not regard other baselines as irrelevant. Originally, we used DQN/Rainbow as baselines because they are standard for Atari games. To strengthen our results, we have included more recent SOTA model-based recurrent algorithms DreamerV2 [2] and DreamerV3 [3] (evaluated at 20M training frames, same training frames as ours) to our list of baselines. We summarize the results in the following table:
>
> | Method   |  BattleZone | DoubleDunk | NameThisGame | Phoenix | Qbert |
> |-|-|-|-|-|-|
> | Our (m=1) |  $35044\pm 986$ | $8.80\pm 2.13$ | $10977\pm 441$ | $5666\pm 54$ | $15313\pm 56$ |
> | Our (m=2) |  $38164\pm 603$ | $17.68\pm 1.15$ | $14308\pm 289$ | $8010\pm 270$ | $15831\pm 146$ |
> | Our (m=4) |  $40098\pm 900$ | $21.17\pm 0.45$ | $18682\pm 580$ | $13593\pm 1036$ | $19697\pm 599$ |
> | Our (m=8) | $39262\pm 763$ | $21.49\pm 0.38$ | $19638\pm 633$ | $18127\pm 1075$ | $23451\pm 415$ |
> | Rainbow | $40061\pm 1866$ | $22.12\pm 0.34$ | $9026\pm 193$ | $8545\pm 1286$ | $17383\pm 543$ |
> | DreamerV2 | $21225\pm743$ | $12.95\pm1.31$ | $12145\pm87$ | $9117\pm2151$ | $3441\pm969$ |
> | DreamerV3 | $37818\pm3979$ | $22.46\pm0.35$ | $20652\pm1002$ | $56889\pm3402$ | $23540\pm1976$ |
>
> We note that DreamerV3 uses a significantly larger model (\~200M parameters) compared to our m=8 model (\~50M parameters). To enable a closer comparison, we also include DreamerV2 (\~20M parameters). The results show that our method (m=8) achieves performance close to DreamerV3 in 3 out of 5 environments, while our smaller models (m=2, \~35M parameters) outperforms DreamerV2 in 4 out of 5 environments. Finally, while we also learn dynamics models for our methods, we differ from Dreamer in that we do not use these models for rollouts. Instead, we leverage them solely for the off-policy corrections in the DAE objective (the B constraint).
>
> > Would it be possible to report ablations on using a Exponential Moving Average network vs a periodically updated network (DQN-style) as your target network? If not would it be possible to justify the choice a bit more?
>
> We have included an additional ablation study regarding EMA vs periodic copy in the latest revision (see Appendix C.5). In summary, we find EMA to outperform periodic copy in 4 out of 5 environments. We believe this is partially due to SPR style self-supervised learning methods being more stable when trained with EMA updates [4],
>
> [1] Agarwal, Rishabh, et al. "Deep reinforcement learning at the edge of the statistical precipice." Advances in neural information processing systems 34 (2021): 29304-29320.
>
> [2] Hafner, Danijar, et al. "Mastering atari with discrete world models." arXiv preprint arXiv:2010.02193 (2020).
>
> [3] Hafner, Danijar, et al. "Mastering diverse domains through world models." arXiv preprint arXiv:2301.04104 (2023).
>
> [4] Ni, Tianwei, et al. "Bridging State and History Representations: Understanding Self-Predictive RL." arXiv preprint arXiv:2401.08898 (2024).

---

> > ### Comment · Reviewer_6QRc · 2024-11-25
> >
> > My complaint wasn't exactly about how SOTA the results were, but rather on the importance of components like the recurrent architecture. In any case the comparison with Dreamerv3 probably implied a favourable comparison against agents like R2D2.
> > Also thank you for the added results comparing the different target network methods. Could you please re-state how many seeds you're using on Figure 10? Most methods seem to end up with 25% of runs being above human normalized score = 4, which seems confusing when you're using 10 seeds according to Figure 3's legend

---

> > > ### Author Response · Authors · 2024-11-25
> > >
> > > Thank you for the reply.
> > >
> > > For Figure 10, we use 10 seeds for our methods, 5 seeds for Rainbow (reported by [1]), 4 seeds for DreamerV2 (reported by [2]), and 6 seeds for DreamerV3 (reported by [3]). The tail at 20% is due to Double Dunk's high human-normalized score across all agents (>12), driven by the low human baseline score of -16.4 in this environment, which heavily skews the curves. We only showed the human-normalized score up to 4, as the scores for other environments are largely within this region.
> > >
> > > We hope this clarifies your questions. Please let us know if there are any remaining concerns.
> > >
> > > [1] https://github.com/google/dopamine/tree/master/baselines/atari
> > >
> > > [2] https://github.com/danijar/dreamerv2
> > >
> > > [3] https://github.com/danijar/dreamerv3

---

> ### Comment · Reviewer_6QRc · 2024-12-02
>
> I agree with Reviewer nsQ7 that further evidence of the importance of partial observability for the algorithm's ability to outperform others being necessary, as such my score will also remain unchanged

---

### Official Review · Reviewer_nsQ7 · 2024-11-04

**Soundness:** 4
**Presentation:** 4
**Contribution:** 2
**Rating:** 6
**Confidence:** 3

**Summary:**

The paper extends and improves an existing off-policy reinforcement learning (RL) algorithm called direct advantage estimation (DAE) to the setting of partially observable environments.

The paper motivates learning advantage functions, as opposed to value functions, given their more stable learning dynamics. Then, it explains how DAE can work in the off-policy case by separating the return in two advantage functions: one associated with the actions of the agent and the other with the transitions, interpreted as environment actions. Based on this separation, the paper re-introduces value and advantage functions as functions of observation (and action and reward) histories, rather than states, and provides an optimization problem whose minima contain the real value and advantage functions.

Moreover, it is explained that, since some of the constraints in the optimization problem depend on having access to the real dynamics, there is a need for some type of approximate model. As a solution, the paper introduces a predictive recurrent architecture that models the dynamics of the environment in an embedding space, mimicking a discrete version of variational autoencoders (VQ-VAEs).

Finally, the paper evaluates the performance of the algorithm based on return maximization in 5 different Atari environments, including some ablations. The results show that the algorithm can compete with state-of-the-art methods like Rainbow with a considerably smaller number of samples, that the off-policy correction introduced by the use of the environment's advantage function is critical to compete with existing techniques, that there are diminishing returns on the multi-step size considered to estimate advantages, that frame-stacking can be replaced by an LSTM given that the algorithm supports partial observability, and that naive truncation of the history can be detrimental.

**Strengths:**

Originality:
- The paper makes use of a simple modification to a previously introduced concept to be able to leverage previous theory. In particular, the paper redefines the environment's advantage function in terms of the reward and V- and Q- value functions, as opposed to just the V-value function. This allows transferring an existing algorithm to the more challenging partially observable setting.
- The paper provides a conceptually simple predictive model in encoding space to manage the approximation of the dynamics.

Quality:
The paper is thorough in its empirical evaluation. In particular, it includes a sufficient number of seeds, relevant baselines, and multiple ablations. Some of the ablations are particularly appropriate since they provide some evidence that the algorithm does maximize return under partial observability.

Clarity:
In general, the paper is easy to read due to its simplicity and clarity. It introduces very succinctly the problem and clearly states the contributions. It also introduces all the relevant notation and explains in detail the experiments carried out.

Significance:
The paper provides convincing evidence to conclude that the proposed algorithm is state-of-the-art. In particular, it is shown that the algorithm makes use of fewer samples to obtain similar or better performance than the baselines in medium size environments that require the use of function approximation and handling some degree of partial observability.

**Weaknesses:**

In my opinion, there is only one weakness worth focusing on. The main contribution is supposed to be the extension of DAE to the partially observable setting, but the Atari environments chosen are arguably not challenging from the partially observable perspective. In particular, the fact that DQN and Rainbow, algorithms that are typically paired with non-recurrent architectures, can maximize rewards in them suggests that partially observability is not a big issue on them. In this sense, I think that ablation studies in these environments, for example, one comparing frame-stacking vs. LSTMs, are not as effective as choosing tasks where partial observability is a crucial factor for good performance.

Now, while I consider that the paper should be accepted in its current version, I think that a higher score requires considering more appropriate environments. For example, see Pašukonis et al., Evaluating Long-Term Memory in 3D Mazes, in ICLR, 2023.

**Questions:**

- What is the computational overhead of the predictive model?
- In line 246, why is $\hat{x}$ being called an information vector?

---

> ### Author Response · Authors · 2024-11-19
>
> We thank the reviewer for the detailed and constructive feedback, please see our answers to your questions below.
>
> > In my opinion, there is only one weakness worth focusing on. The main contribution is supposed to be the extension of DAE to the partially observable setting, but the Atari environments chosen are arguably not challenging from the partially observable perspective. In particular, the fact that DQN and Rainbow, algorithms that are typically paired with non-recurrent architectures, can maximize rewards in them suggests that partially observability is not a big issue on them. In this sense, I think that ablation studies in these environments, for example, one comparing frame-stacking vs. LSTMs, are not as effective as choosing tasks where partial observability is a crucial factor for good performance.
> >
> > Now, while I consider that the paper should be accepted in its current version, I think that a higher score requires considering more appropriate environments. For example, see Pašukonis et al., Evaluating Long-Term Memory in 3D Mazes, in ICLR, 2023.
>
> We would like to note that while agents with non-recurrent architectures (such as Rainbow) have achieved strong performance in Atari games, previous research has demonstrated that recurrent architectures can significantly improve performance across various environments (e.g., R2D2 [1]). This strongly suggests that partial observability in Atari games is a non-trivial factor. As a more concrete example, in Battle Zone, players control a tank to eliminate enemies and avoid attacks while having a limited view of its surrounding at each time step, which aligns closely with the definition of partial observability.
>
> To the best of our knowledge, the extent of partial observability in Atari remains an open question. We believe our ablation studies provide valuable evidence supporting the presence of partial observability in these games, while also offering new insights, such as the confounding bias, to this problem.
>
> > What is the computational overhead of the predictive model?
>
> For our models, the computational overhead of the predictive model, which consists of 3 MLPs (each with a single hidden layer), is similar to the value heads ($\hat{A}, \hat{B}, \hat{V}$) and almost negligible compared to other parts (convolutional layers or recurrent layers) of the network.
>
> > In line 246, why is $\hat{x}$ being called an information vector?
>
> We have corrected this typo and made the notations more consistent in the latest revision.
>
> [1] Kapturowski, Steven, et al. "Recurrent experience replay in distributed reinforcement learning." International conference on learning representations. 2018.

---

> > ### Comment · Reviewer_nsQ7 · 2024-11-24
> >
> > Thanks for your reply, but I have to disagree with your answer. The fact that there is partial observability does not mean that it is a crucial factor for good performance. This is precisely proved by Rainbow performing on par or better than the rest of the algorithms. Hence, I do not see any strong evidence suggesting that the reason why the proposed algorithm is competitive is because it deals with partial observability.
> >
> > Given no additional empirical results focused on partial observability, my score remains unchanged.

---

### Author Response · Authors · 2024-11-19
**General response**

We thank the reviewers for the valuable feedback and thoughtful comments, which have greatly helped improve the quality of this work. Here, we summarize the changes:

1. **Baselines**: As pointed out by the reviewers, the baselines (DQN/Rainbow) used in this paper, while standard for the ALE, are no longer considered SOTA anymore. Consequently, we have updated our list of baselines to include more recent SOTA model-based recurrent algorithms DreamerV2 [1] and DreamerV3 [2].
2. **Notations**: As noted by the reviewers, there were some inconsistencies in the usage of state variables ($s$) and information vectors ($h$), which could be confusing. To address this, we have updated the notations in Section 3 to ensure better consistency across different functions. Additionally, we have revised Figure 1 to more clearly illustrate the connections between the variables ($o_t, x_t, h_t$) and the VQ-VAE.
3. **New ablation**: As suggested by reviewer 6QRc, we have added a new ablation study (see Appendix C.5) that compares periodic updates of the target, common in DQN variants, to EMA-style updates, more typical in self-supervised methods like SPR.

We thank the reviewers once again and hope that our responses, along with the revised manuscript, address any confusion and resolve all outstanding issues.

[1] Hafner, Danijar, et al. "Mastering atari with discrete world models." arXiv preprint arXiv:2010.02193 (2020).

[2] Hafner, Danijar, et al. "Mastering diverse domains through world models." arXiv preprint arXiv:2301.04104 (2023).

---

> ### Author Response · Authors · 2024-11-24
> **Additional experiment with DreamerV3**
>
> As the DreamerV3 scores reported in [1] are based on a much bigger model (~200M parameters) compared to ours (m=8, ~50M parameters), we conducted additional experiments using the 50M-parameter variant of DreamerV3 (see table 3 in [1]) for a closer comparison. In addition to model capacity, we also examine the impact of replay ratio, which specifies how much training was done (see page 18 of [1] for further details). All DreamerV3 experiments were based on the official implementation provided in [2] and were repeated over 3 random seeds.
>
> The following table summarizes the result after 20M training frames (RR: replay ratio, values represent mean$\pm$1 standard error):
> | Method   |  BattleZone | DoubleDunk | NameThisGame | Phoenix | Qbert |
> |-|-|-|-|-|-|
> | Our (m=8, RR=16) | $39262\pm 763$ | $21.49\pm 0.38$ | $19638\pm 633$ | $18127\pm 1075$ | $23451\pm 415$ |
> | DreamerV3 (RR=16) | $6333\pm3383$ | $19.33\pm2.67$ | $14950\pm1139$ | $11297\pm1832$ | $17883\pm1295$ |
> | DreamerV3 (RR=32) | $23667\pm2906$ | $22.67\pm0.67$ | $15640\pm859$ | $24307\pm1813$ | $22808\pm841$ |
>
> (by default, DreamerV3 uses replay ratio = 32)
>
> In general, we find our method outperforms DreamerV3 (RR=16), and stays competitive with DreamerV3 (RR=32).
>
> We hope these additional experiments address any questions regarding the baselines, and thank the reviewers once again for their constructive feedback.
>
> [1] Hafner, Danijar, et al. "Mastering diverse domains through world models." arXiv preprint arXiv:2301.04104 (2023).
>
> [2] https://github.com/danijar/dreamerv3

---

### Meta-Review · Area_Chair_y9Tj · 2024-12-18

**Metareview:**

This paper extends prior work on off-policy Direct Advantage Estimation (DAE) by enabling support for partially-observable systems. The authors provide theoretical justification for their approach and couple it with an empirical evaluation on the ALE.

All reviewers agree that the paper is clearly written (especially after revisions during the rebuttal), the method interesting, and the theory well presented.

However, given how heavily the proposed method builds on DAE with the claim that this enables running on partially observable environments, it was felt that the provided empirical evidence is not sufficient to fully justify its need. In particular, all reviewers agreed that more games on the ALE would be useful. Although the five were chosen based on the suggestion of the Atari-5 paper, it is not clear that those 5 are the best for highlighting issues with partial observability, which is the main focus of this work.

Perhaps showcasing this method in a setting where the distinction between full and partial observability (beyond frame-stacking) would clarify the distinction.

In summary, this is interesting work that unfortunately does not quite yet meet the bar for publication, for the reasons mentioned above.

**Additional Comments On Reviewer Discussion:**

Most of the points raised by the reviewers were with regards to the empirical evaluations, as well as some issues with clarity.

The authors addressed all the clarity issues and did run extra experiments with Dreamer-v2 and -v3. However, the main issue remains regarding properly evaluating the need for the proposed method in partially observable settings.

---

### Decision · Program_Chairs · 2025-01-22

Reject